# CALIPSO lidar level 3 aerosol profile product: version 3 algorithm design

Jason L. Tackett[1,2], David M. Winker[2], Brian J. Getzewich[1,2], Mark A. Vaughan[2], Stuart A. Young[1], and Jayanta Kar[1,2]

[1]Science Systems and Applications, Inc., Hampton, VA, USA
[2]NASA Langley Research Center, Hampton, VA, USA

*Correspondence to*: Jason L. Tackett (jason.l.tackett@nasa.gov)

**Abstract.** The CALIPSO (Cloud-Aerosol Lidar and Infrared Pathfinder Satellite Observations) level 3 aerosol profile product reports globally gridded, quality-screened, monthly mean aerosol extinction profiles retrieved by CALIOP (the Cloud-Aerosol Lidar with Orthogonal Polarization). This paper describes the quality screening and averaging methods used to generate the version 3 product. The fundamental input data are CALIOP level 2 aerosol extinction profiles and layer classification information (aerosol, cloud, and clear-air). Prior to aggregation, the extinction profiles are quality-screened by a series of filters to reduce the impact of layer detection errors, layer classification errors, extinction retrieval errors, and biases due to an intermittent signal anomaly at the surface. The relative influence of these filters are compared in terms of sample rejection frequency, mean extinction, and mean aerosol optical depth (AOD). The "extinction QC flag" filter is the most influential in preventing high-biases in level 3 mean extinction, while the "misclassified cirrus fringe" filter is most aggressive at rejecting cirrus misclassified as aerosol. The impact of quality screening on monthly mean aerosol extinction is investigated globally and regionally. After applying quality filters, the level 3 algorithm calculates monthly mean AOD by vertically integrating the monthly mean quality-screened aerosol extinction profile. Calculating monthly mean AOD by integrating the monthly mean extinction profile prevents a low bias that would result from alternately integrating the set of extinction profiles first and then averaging the resultant AOD values together. Ultimately, the quality filters reduce level 3 mean AOD by −24 and −31 % for global ocean and global land, respectively, indicating the importance of quality screening.

## 1 Introduction

In October 2015 the CALIPSO (Cloud-Aerosol Lidar and Infrared Pathfinder Satellite Observation) team released the version 3 level 3 aerosol profile product, based on aerosol extinction retrievals from the spaceborne elastic backscatter lidar, CALIOP (i.e., the Cloud-Aerosol Lidar with Orthogonal Polarization). Version 3 was the first official release, replacing the beta version released in 2011 and described in Winker et al. (2013). Summarizing more than 10 years of retrievals, the level 3 aerosol profile product contains a near-global (82° S–82° N) record of quality-screened aerosol extinction profiles and aerosol optical depth (AOD), reported as monthly averages on a uniform 2° latitude by 5° longitude grid. Currently, CALIOP provides the longest record of the vertical distribution of tropospheric aerosol occurrence,

extinction, and speciation. Given the uniqueness of the dataset, the level 3 aerosol profile product has been embraced by the scientific community for a variety of applications.

Researchers have used the CALIOP level 3 product to investigate seasonal variability of the vertical distribution and extinction profiles (Huang et al., 2013; Xu et al., 2015). It has provided insights into global aerosol source attribution (Prijith et al., 2013) and how the vertical distribution of aerosols relates to atmospheric circulation (Alizadeh-Choobari et al., 2014; Prijith et al., 2016) and to ice cloud nucleation potential (Tan et al., 2014). Vertical extinction profiles have helped to interpret seasonal surface $PM_{2.5}$ variability (Ma et al., 2016) and to evaluate estimates of wildfire injection heights (Sofiev et al., 2013). Aerosol radiative effect investigations have also benefited from the level 3 aerosol product (Adebiyi et al., 2015; Chung et al., 2016).

Over the years, researchers have used various quality screening methods for level 2 aerosol products, sometimes in collaboration with CALIPSO algorithm developers (Kittaka et al., 2011; Campbell et al., 2012a; Koffi et al., 2012; Redemann et al., 2012; Toth et al., 2013; Kacenelenbogen et al., 2014). These quality screening methods were similar to those used to generate the level 3 aerosol product. Quality screening procedures for the beta level 3 aerosol product were initially reported by Winker et al. (2013). In subsequent years, researchers have adopted these procedures explicitly (Sarangi et al., 2016; Marinou et al., 2017) while others have adopted variations on these procedures, citing the level 3 aerosol product as a reference (Ge et al., 2014; Todd and Cavazos-Guerra, 2016).

This paper documents the averaging and quality screening methods used to generate the version 3 level 3 aerosol profile product. The goal is to aid the community's understanding of the product and provide guidance for the use of CALIOP aerosol data. Validation is not reported since validating level 3 aerosol extinction profiles against independent observations necessarily involves validating level 2 layer detection, lidar ratio selection, and extinction retrievals. Given the breadth of these tasks, validation of the level 3 aerosol product will be reported in a future publication.

This paper is organized as follows: First, a summary of the CALIPSO level 2 algorithms relevant to the level 3 aerosol product is given in Sect. 2. An overview of the level 3 product structure and contents is given in Sect. 3. Methods for averaging extinction and computing AOD are described in Sect. 4. Quality screening procedures are detailed in Sect. 5. Overall impact of quality screening on quantities reported by the level 3 aerosol product is discussed in Sect. 6, prior to the summary given in Sect. 7. Additional figures are reported in supplemental material.

## 2 CALIOP overview and level 2 aerosol product descriptions

The CALIPSO satellite has been observing the vertical distribution of aerosols and clouds since June 2006. The primary instrument on CALIPSO is CALIOP, a nadir-viewing dual-wavelength (532 and 1064 nm), dual-polarization (at 532 nm), elastic backscatter lidar (Hunt et al., 2009). CALIOP measures profiles of attenuated backscatter from the Earth's atmosphere and surface every 333 m along the orbit track, which are reported in the level 1B data product.

Level 2 algorithms then detect features, assign type classifications (aerosol, cloud, surface), and retrieve extinction coefficients from the attenuated backscatter signals. Features are detected in the atmosphere using a multi-resolution averaging engine with altitude-dependent thresholds that optimize compromises between spatial resolution and signal-to-noise ratio (Vaughan et al., 2009). Both strongly scattering and weakly scattering features are detected by averaging level 1B profiles, having a fundamental spatial sampling of 1/3 km horizontally, to multiple coarser resolutions (5, 20, and 80 km). Features detected at higher resolution are removed prior to averaging to coarser resolutions to allow successively fainter features to be detected. Once a feature is detected, it is stored as a "layer", having specific top and base altitudes, and a horizontal extent based on the averaging required for detection. A cloud-aerosol-discrimination (CAD) algorithm then determines the feature type (aerosol, cloud, or stratospheric feature) by evaluating selected spatial and optical properties of the layer against a five-dimensional probability density function (Liu et al., 2009). In the version 3 level 2 algorithms, all layers detected at 1/3 km resolution are, by default, classified as clouds. Also in version 3, layers detected above the tropopause are classified only as "stratospheric features" rather than aerosol or cloud.

To calculate extinction coefficients, the extinction retrieval algorithm requires a lidar ratio (i.e., the ratio of extinction to backscatter) for the layer being analyzed. Lidar ratios are either selected based on the layer type or derived iteratively from the measured layer transmittance (Young and Vaughan, 2009). Derived lidar ratios, obtained from these "constrained retrievals", are rarely obtained for aerosols (< 0.01 % of all aerosol layers detected). Most often, aerosol lidar ratio selection relies on an aerosol subtyping algorithm to classify the aerosol as one of six subtypes: clean marine, dust, polluted dust, clean continental, polluted continental, or smoke (Omar et al., 2009). Each of these aerosol subtypes is assigned a default lidar ratio derived from a combination of AERONET cluster analysis, theoretical scattering calculations, and direct measurements (Omar et al., 2009).

The extinction algorithm retrieves vertical profiles of extinction, reported separately for aerosols and clouds. Aerosol extinction is not reported within clouds because the lidar signals are dominated by cloud scattering and so atmospheric features are classified as either aerosol or cloud and the retrieved extinction is reported for only one or the other. Another fundamental feature of the level 2 algorithms is that extinction is only reported for detected features; i.e., extinction is not retrieved in regions classified in level 2 as "clear-air" although there may be aerosol below the detection limit (Sect. 4.1). Retrieved and measured quantities for detected aerosols are used to construct two different level 2 aerosol products: an aerosol layer product and an aerosol profile product. The aerosol layer product reports layer-averaged and layer-integrated quantities. The aerosol profile product combines the profiles retrieved within (possibly overlapping) aerosol layers to report vertical profiles of extinction coefficients, layer detection information, and quality assurance parameters at 5 km horizontal resolution. The vertical resolution is 60 m from −0.5 km to 20.2 km, and 180 m above 20.2 km.

The level 3 aerosol profile product is derived from the level 2 aerosol profile product. This is a fundamental point because alternately using the level 2 aerosol layer product can misrepresent the shape of the aerosol extinction profile. For instance, extinction profiles can also be estimated from the layer product by assuming the aerosol is vertically distributed uniformly within the layer. Layers can be several kilometers deep, however, and this assumption can significantly distort the

estimated shape of the extinction profile. This is illustrated schematically in Fig. 1(a), where the red dashed line indicates the layer-averaged extinction value. Conversely, the blue solid line illustrates how the extinction profile might look as reported in the profile product. In this example, using the aerosol layer product would underestimate aerosol extinction at low altitudes and overestimate extinction at high altitudes. These over/underestimates are also evident in Fig. 1(b), which uses

5    CALIOP level 2 data. Here, seasonal mean aerosol extinction profiles are computed over the central tropical Atlantic from the layer product using layer-average extinction and from the profile product using the reported aerosol extinction profiles. This region is characterized by an inversion layer at about 2 km with transported Sahara dust above and primarily marine aerosol below. As in the schematic example, aerosol extinction is underestimated below 1 km and overestimated at higher altitudes. In order to capture the extinction profile shape as retrieved by CALIOP, the level 2 profile product must be used.

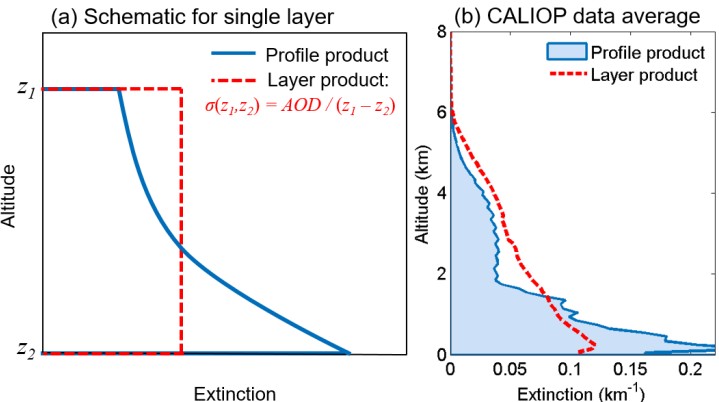

**Figure 1. (a) Schematic example of aerosol extinction profile for an individual layer reported by the profile product (blue) and by the layer product (red dashed) where mean aerosol extinction is computed from the layer AOD divided by the geometric depth (Δz). (b) Seasonal average of CALIOP aerosol extinction computed by the profile product (blue, filled) and by the layer product (red dashed) for June – August 2007 at night over the central Atlantic Ocean region (Table A1).**

15         The following nomenclature is used throughout the remainder of this paper. "Columns" are 5 km horizontal averages along the CALIPSO orbit track (i.e., 15 consecutive level 1B profiles). "Layers" are features detected by the CALIOP feature finder. Within the level 2 processing, extinction profiles are only retrieved for those layers detected at horizontal averages of 5 km, 20 km and 80 km. Layers therefore span one, four or sixteen columns, according to the averaging required for detection. Layers are unique entities, regardless of the number of columns they span. "Samples" refer

20    to individual range bins within the level 2 profile product (e.g., a layer can have multiple aerosol extinction samples within its vertical extent).

## 3 Level 3 aerosol profile product overview

        The CALIOP level 3 aerosol profile product reports monthly statistics based on quality-screened level 2 aerosol extinction profiles at 532 nm below 12 km in altitude, vertically gridded with respect to mean sea level. Profiles are reported

near-globally (85° S to 85° N) on a uniform 2° latitude by 5° longitude grid with a vertical resolution of 60 m. The 12 km upper limit was selected due to the rarity of tropospheric aerosol detection above 12 km in the level 2 product (e.g., 0.04 % of tropospheric aerosol layers detected by CALIOP version 3 are above 12 km in 2010). The focus is therefore on the lower troposphere. Eight level 3 files are generated for each month: day and night files for each of four different sky conditions:

all-sky, cloud-free, cloudy-sky transparent, and cloudy-sky opaque. Figure 2 depicts these sky conditions for an individual level 2 granule. White areas in Fig. 2 are excluded for the given sky condition, defined below:

  • "All-sky" averages are constructed from all quality-screened aerosol extinction coefficients, regardless of cloud cover.

  • "Cloud-free" averages are constructed from columns where no clouds are detected at 5 km or coarser horizontal

resolution. Boundary layer clouds detected at 1/3 km are removed by the level 2 boundary layer cloud-clearing algorithm prior to averaging the attenuated backscatter and retrieving extinction.

  • "Cloudy-sky transparent" averages are constructed from columns containing clouds detected at 5 km or coarser resolution where the surface is still detected; i.e., the CALIOP signals reach the Earth surface and the profile contains clouds. Aerosol layers may lie above or below the clouds.

• "Cloudy-sky opaque" averages are constructed from columns containing clouds detected at 5 km or coarser resolution where the surface is not detected because the lowest cloud layer is opaque. Only level 2 aerosol extinction from 12 km in altitude down to the top of the opaque cloud contribute to the average. By definition, sampling for both cloudy sky conditions is dependent on cloud cover.

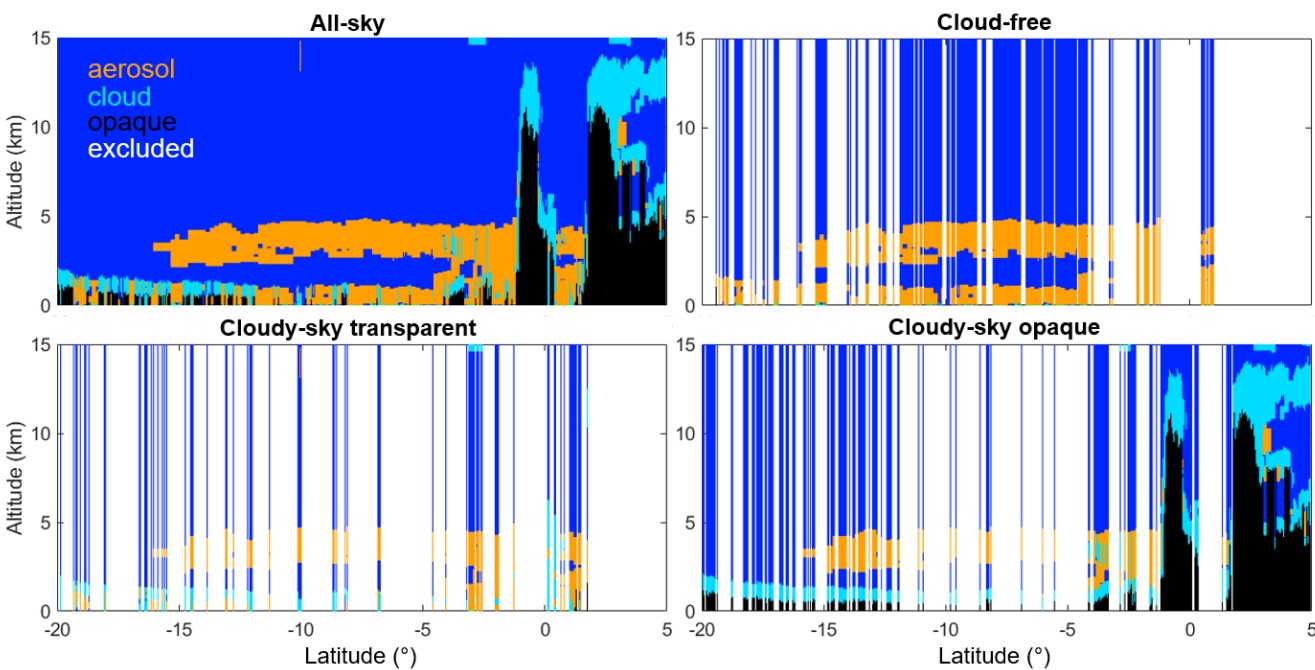

**Figure 2. Feature classifications for an individual nighttime level 2 granule (2008-01-01T01-30-23ZN) demonstrating the four level 3 sky conditions. Data in white columns are excluded for the indicated sky condition. Clouds, aerosols and totally attenuated (opaque) features are light blue, orange, and black, respectively.**

Separating level 3 files into four different sky conditions based on cloud cover has several important benefits. All-sky provides the greatest sampling of all the sky conditions, thereby providing the most information about aerosol extinction within the atmosphere. The cloud-free sky condition represents the highest quality level 3 data since extinction retrievals are minimally affected by errors in retrieving the attenuation of overlying cloud cover. Further, the daytime cloud-free sky condition provides sampling similar to aerosol products from MODIS (Moderate Resolution Imaging Spectroradiometer) and other passive remote sensors in which aerosol observations are reported for cloud-free skies. The CALIOP cloud mask, however, is quite different than the MODIS cloud mask and reports higher global mean cloud cover because of CALIOP's ability to detect subvisible cirrus (Stubenrauch et al., 2013). Statistics from cloudy-sky transparent files can be aggregated with cloud-free statistics to increase sampling, although the former sky condition is expected to have larger uncertainties. The cloudy-sky opaque sky condition primarily reports aerosol above low water clouds. Note that the cloud-free, cloudy-sky transparent, and cloudy-sky opaque sky conditions are disjoint sets. When weighted by the number of samples averaged (Sect. 4.3), the mean extinction for these sky conditions sum to the all-sky mean extinction.

Daytime and nighttime retrievals are reported in separate level 3 files because measurement noise and layer detection sensitivities are different. In daytime, the signal-to-noise ratio (SNR) is lower relative to night, particularly over high albedo surfaces such as desert or snow or over clouds (Hunt et al., 2009). This reduces the ability to detect faint layers that would otherwise be detectable at night (Winker et al., 2013). Lower SNR also contributes to higher uncertainty in the daytime level 2 extinction retrievals (Young et al., 2013). Separating day and night retrievals into separate files avoids mixing disparate levels of uncertainty and layer detection capability.

The primary data sets in the level 3 aerosol profile product – and the focus of this paper – are vertical profiles of mean aerosol extinction and mean AOD at 532 nm. These quantities are reported for all aerosol species together and for the following individual aerosol species: dust, polluted dust, and smoke. In addition, sampling statistics are included, which fully account for the disposition of every level 2 sample evaluated by the level 3 algorithm. During the quality screening and averaging process, samples in the level 2 aerosol extinction array are either accepted, rejected, ignored, or excluded. Sampling statistics document this information along with the number of samples contributing to the average and the total number of samples searched. Samples described in this paper as rejected, ignored, or excluded do not contribute to the mean extinction calculation. Ignored samples contribute to the number of samples searched whereas excluded samples do not (e.g., cloud and stratospheric features are ignored while opaque, surface, and subsurface features are excluded).

Operationally, the level 3 algorithm iterates through all level 2 files within a month. Aerosol extinction samples are quality-screened and then aggregated along with their sampling statistics into appropriate latitude, longitude grid cells. Once all level 2 files are evaluated, the quality-screened extinction profiles are averaged and integrated for each grid cell. The following section describes the procedures for averaging and integration, dedicating the remainder of the paper (Sects. 5–6)

to describing the quality screening strategy. Table 1 is given here as a high-level summary of the averaging methods and quality filtering procedures detailed in the following two sections.

**Table 1. Summary of averaging methods and quality filtering procedures used to generate the version 3 level 3 aerosol product. Details are discussed in the indicated sections. AGL and AMSL indicate "above ground level" and "above mean sea level", respectively.**

| Averaging method / quality filtering procedure | Section |
| --- | --- |
| Aerosol extinction for "clear-air" assigned $\equiv$ 0 km$^{-1}$ | 4.1 |
| Clear-air below aerosol layers with bases < 250 m AGL ignored | 4.2 |
| Isolated 80 km horizontal resolution aerosol layers rejected | 5.1 |
| CAD score outside [−100, −20] range rejected | 5.2.1 |
| Aerosol in contact with ice clouds (top temperature < 0° C) above 4 km AMSL rejected | 5.2.2 |
| Extinction QC flag $\neq$ 0, 1, 16, 18 rejected | 5.3.1 |
| Extinction uncertainty = 99.9 km$^{-1}$ rejected, and all extinction below | 5.3.2 |
| All samples $\leq$ 60 m AGL excluded | 5.4 |

## 4 Averaging and integration methods

This section describes the averaging and integration methods employed to produce profiles of mean aerosol extinction and mean AOD following quality screening (described in Sect. 5). The first task is to account for aerosol extinction within "clear-air" range bins where features have not been detected. Next, a mitigation strategy is described that avoids low biases in mean level 3 aerosol extinction caused when aerosol is undetected at the bases of surface-attached aerosol layers. Finally, the mathematics of averaging and integration are presented.

### 4.1 Aerosol in "clear-air" regions

Aerosol extinction is only retrieved where aerosol is detected by the CALIOP feature finder. Level 2 atmospheric samples classified as "clear-air" (i.e., no feature is detected) are assumed in the level 3 algorithm to have aerosol extinction equal to 0 km$^{-1}$, denoted by $\sigma_{clear}$ (specifically, extinction $\sigma$ for clear-air samples are assigned $\sigma \equiv \sigma_{clear}$ ; the triple bar denotes the assignment). However, because layer detection is based on vertically resolved backscatter, diffuse aerosol layers which span a large altitude range can remain undetected, particularly if they have significant absorption. Solar background noise further impacts feature detection (Winker et al., 2013). Assuming $\sigma_{clear} = 0$ km$^{-1}$ thereby provides a lower bound on the true aerosol extinction. In reality, aerosol is present virtually everywhere throughout the troposphere (e.g., Kim et al., 2017), though concentrations can be very low in regions of the free troposphere not affected by continental transport. Clarke and Kapustin (2002), for example, show background aerosol extinction levels of 10$^{-4}$ km$^{-1}$ to 10$^{-3}$ km$^{-1}$ in remote parts of the

Pacific basin, implying a missing AOD ranging from $10^{-3}$ to $10^{-2}$ in the cleanest regions (assuming well-mixed aerosols in a 10 km deep column).

Several researchers have recently sought to characterize the optical depths of the aerosol layers undetected by CALIOP using collocated observations (Kacenelenbogen et al., 2011; Sheridan et al., 2012; Rogers et al., 2014; Thorsen and Fu, 2015; Toth et al., 2018) or independent retrievals (Winker et al., 2013; Kim et al., 2017). Exactly how these undetected layers affect the level 3 mean extinction is difficult to estimate given that the resulting underestimate depends on the magnitude of missing extinction and the frequency of non-detection. Answering this question is a topic for forthcoming level 3 aerosol product validation.

### 4.2 Undetected near-surface aerosol

The CALIOP feature finder sometimes leaves a gap between the base of the lowest aerosol layer and the surface, even in cases where the aerosol layer extends to the surface. An example over the Pacific Ocean is shown in Fig. 3(a), circled in red. In this region, the dominant aerosol source is the ocean itself and the marine boundary layer is well-mixed, so it is reasonable to expect aerosol to exist down to the surface. However, aerosol is not identified in range bins near the surface. The level 2 aerosol base extension algorithm is designed to compensate for situations like this by extending aerosol layer bases downward to capture more of the surface-attached layer (Vaughan et al., 2010). However, gaps of apparent clear-air between the surface and aerosol layer base can remain for two reasons. First, base extension is only executed if the integrated attenuated backscatter signal between the original layer base and the surface is positive. In Fig. 3(b) the backscatter signal adjacent to the surface is strongly negative due to the negative signal anomaly (discussed in Sect. 5.4) so these aerosol layer bases are not extended. Second, the base extension algorithm only extends layer bases to 90 m above the local surface in order to prevent the surface signal from contaminating the aerosol profile. These "clear-air" gaps would cause a low-bias in the level 3 mean aerosol extinction profile near the surface if they were assigned $\sigma_{clear} = 0 \text{ km}^{-1}$.

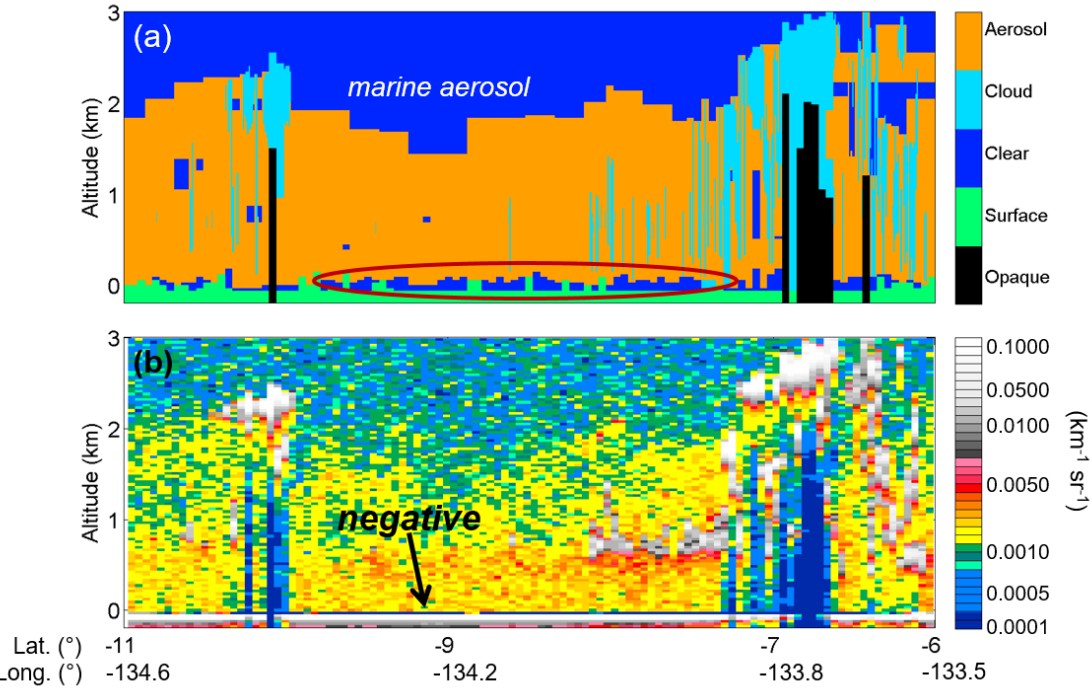

**Figure 3. (a) Level 2 feature type classification and (b) level 1B total attenuated backscatter for the granule 2008-08-01T10-17-21ZN passing over the Pacific Ocean. Undetected surface-attached aerosol (circled) and negative attenuated backscatter (arrow) are denoted.**

To avoid a low bias in near-surface mean aerosol extinction, the level 3 algorithm ignores all clear-air samples below the lowest aerosol layer in each column having a base below 250 m. The underlying assumption is that the atmosphere is well mixed below 250 m. Turbulent mixing within the daytime boundary layer tends to homogenize aerosol loading, and the planetary boundary layer is generally much deeper than 250 m for marine and continental conditions (e.g., McGrath-Spangler and Denning (2013); Luo et al. (2014)). [Note that the beta version of the level 3 product used 2.46 km rather than 250 m as the threshold (Winker et al., 2013)]. Ignoring the range bins in near-surface gaps gives more weight to range bins where aerosol was detected, preventing a low-biased level 3 average. Figure 4 shows the effect on a level 3 mean aerosol extinction profile over the Arabian Sea. The extinction of the range bin nearest to the surface is increased, making the drop-off in extinction less severe. Consequently, global mean level 3 AOD is increased by a small amount, roughly 1 %.

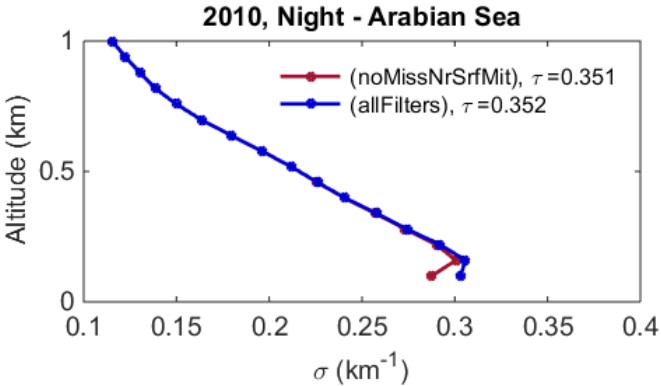

**Figure 4. Mean aerosol extinction with (blue) and without (red) undetected near-surface aerosol mitigation over the Arabian Sea [11° N, 27° N; 55° E, 70° E], all-sky 2010 at night.**

## 4.3 Averaging method

Mean aerosol extinction is calculated from all quality-screened level 2 aerosol extinction coefficients ($\sigma$) and clear-air samples within each latitude, longitude, altitude grid cell using Eq. (1).

$$\bar{\sigma} = \frac{\sum\limits_{i=1}^{N_{aer}} \sigma_{aer,i} + \sum\limits_{j=1}^{N_{clear}} \sigma_{clear,j}}{N_{aer} + N_{clear}} \tag{1}$$

Here, $\bar{\sigma}$ is the monthly mean aerosol extinction coefficient, $\sigma_{aer,i}$ is the set of aerosol extinction coefficients accepted by quality screening, $\sigma_{clear,j}$ is the set of clear-air aerosol extinction coefficients retained after accounting for near-surface aerosol (Sect. 4.2), $N_{aer}$ is the total number of aerosol extinction samples accepted, and $N_{clear}$ is the number of clear-air samples in the grid cell. Under the assumption that $\sigma_{clear} = 0$ km$^{-1}$ and the definition $N_{avg} = N_{aer} + N_{clear}$, Eq. (1) reduces to Eq. (2).

$$\bar{\sigma} = \frac{\sum\limits_{i=1}^{N_{aer}} \sigma_{aer,i}}{N_{avg}} \tag{2}$$

Profiles of $\bar{\sigma}$, $N_{aer}$ and $N_{avg}$ are reported in the level 3 product with the science data set (SDS) names Extinction_532_Mean, Samples_Aerosol_Detected_Accepted, and Samples_Averaged, respectively. Multi-month averages of aerosol extinction can be calculated from $\bar{\sigma}$ by weighting each month by $N_{avg}$.

Mean aerosol extinction is reported for all aerosol species combined and also reported separately for dust, polluted dust, and smoke. When computing $\bar{\sigma}$ for a single-species, $\sigma_{aer}$ for all other aerosol species is assumed to equal 0 km$^{-1}$. This is consistent with the CALIPSO aerosol typing paradigm where aerosol layers are assigned a single type. In reality, different aerosol types can be mixed within the same layer, but the CALIPSO aerosol typing algorithm is unable to determine when

different species are mixed or by what proportions. Therefore, assigning $\sigma_{aer} \equiv 0$ km$^{-1}$ for other species is equivalent to assuming that only one aerosol type is present in the detected layer. By contrast, the beta version of the level 3 product ignored other species rather than setting their extinction to 0 km$^{-1}$. This caused extinction to be biased high where multiple aerosol subtypes exist at the same altitude, as demonstrated by Amiridis et al. (2013) (their Fig. 7 and accompanying discussion). Assigning $\sigma_{aer} \equiv 0$ km$^{-1}$ for other species avoids these biases and maintains consistency with the CALIPSO aerosol typing paradigm.

## 4.4 Mean AOD calculation

AOD is the standard parameter used by spaceborne passive sensors and sun photometers to quantify total column aerosol loading in cloud-free sky conditions. Temporal averaging is accomplished by averaging a set of AOD measurements/retrievals over the time period of interest. Computing temporally averaged AOD for CALIOP retrievals, however, requires a different approach because the averaging set consists of $\sigma$ profiles rather than total column AOD measurements. In the level 3 product, monthly mean AOD is computed by first averaging the set of quality-screened $\sigma$ profiles for the month and then vertically integrating the mean extinction profile $\bar{\sigma}$, i.e., average-then-integrate. The alternate method is to first integrate each of the quality-screened $\sigma$ profiles and then average the set of AODs, i.e., integrate-then-average. These two methods do not produce the same results. Figure 5 shows that the monthly mean AOD for these two methods is very different for both the all-sky and cloud-free sky conditions; mean AOD is often smaller when integrate-then-average is used. This is because the $\sigma$ profiles in the averaging set do not uniformly sample the same geometric depth of the atmosphere after cloud-clearing and quality screening.

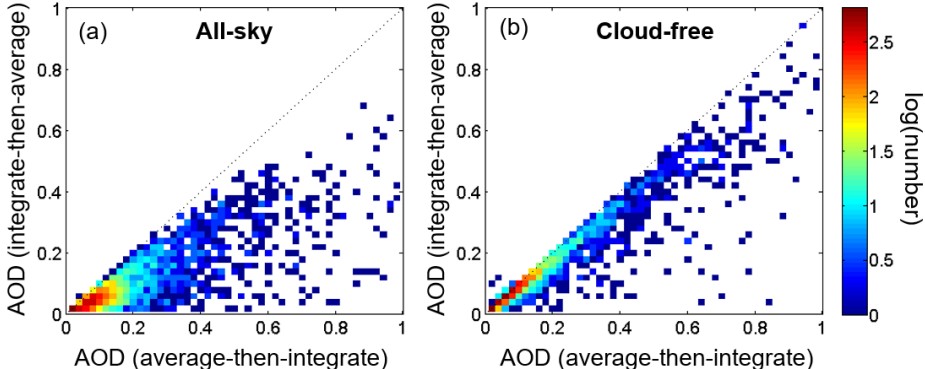

**Figure 5. Level 3 mean AOD for all latitude/longitude grid cells in July 2007 at night for (a) all-sky and (b) cloud-free sky conditions. Colors represent the number of grid cells on a logarithmic scale.**

As a simplified example, the integrate-then-average AOD will be artificially small for two columns where one $\sigma$ profile extends to the surface and the other profile stops at 10 km due to an opaque cloud. The first profile will have a larger AOD because it observed aerosol down to the surface, whereas the second profile will have a smaller AOD because aerosol

observations are terminated at 10 km. These two $\sigma$ profiles do not measure the same geometric depth and the subsequent mean AOD is biased low. This example readily illustrates the mean AOD differences for the all-sky condition where clouds exist in the averaging set (Fig. 5(a)). Further, the cloud-free sky condition also exhibits lower mean AOD for integrate-then-average even though the observations are unencumbered by clouds (Fig. 5(b)). In this case, the geometric depth still differs

between the two methods because $\sigma$ samples are rejected from various range bins by quality screening. The net effect yields $\sigma$ profiles with disparate geometric depths for both the cloud-free and all-sky sky conditions. In short, mean AOD will always be biased low when computed by the integrate-then-average method. Hence, level 3 mean AOD is computed by averaging then integrating. This is an important consideration for computing AOD from space-based profiling instruments.

## 5 Quality screening

CALIOP level 2 data contain many flags and data quality metrics allowing users to screen data to their desired quality level. A number of quality filters are implemented in the level 3 algorithm to prevent untrustworthy level 2 data from contributing to the monthly average (Table 1). These filters are designed to counteract four main issues: noise misclassified as aerosol (Sect. 5.1), clouds misclassified as aerosol (Sect. 5.2), extinction retrieval errors (Sect. 5.3), and an instrument artifact that intermittently produces large negative signals near the surface (Sect. 5.4). All of these filters, except the last, are

identical to filters A1 – A5 described in Appendix A of Winker et al. (2013) for the beta level 3 product. The near-surface negative signal anomaly filter (Sect. 5.4) replaces filter A6 of Winker et al. (2013). Overall, quality filters are applied conservatively. That is, obviously erroneous layers and extinction retrievals are rejected while affecting the $\bar{\sigma}$ profile by the smallest amount possible. A conservative strategy is adopted because, as will be shown, aggressive screening can easily alter not only the magnitude of average extinction, but may also change the $\bar{\sigma}$ profile shape in complex ways. Changing the

profile shape through aggressive quality screening is undesirable because it would cause inconsistencies with level 2 extinction profiles computed by the CALIOP extinction retrieval algorithm whose behavior is relatively well understood. The degree to which these aerosol extinction profile shapes reflect reality (level 2 or level 3) will be addressed in forthcoming validation work.

  This section describes the individual quality filters and demonstrates their impact on the number of aerosol samples

retained after quality screening. Each filter is applied independently, whereas the impact of all filters applied together is examined in Sect. 6. An evaluation period spanning ten years is used (2007–2016). Unless otherwise noted, all statistics refer to nighttime, all-sky for this time period. For context, Figs. 6 and 7 report the total number of aerosol samples prior to quality screening. The frequency of aerosol samples rejected out of all aerosol detected is reported for each individual filter in Figs. 8 and 9. The following subsections will reference these figures significantly. Commensurate daytime figures are reported in

supplementary material as Figs. S1–S4.

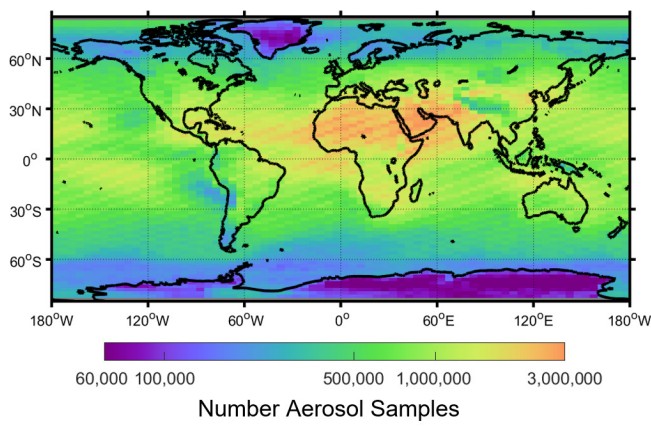

**Figure 6. Total number of aerosol samples reported by the level 3 product prior to quality screening for 2007–2016 at night, all-sky.**

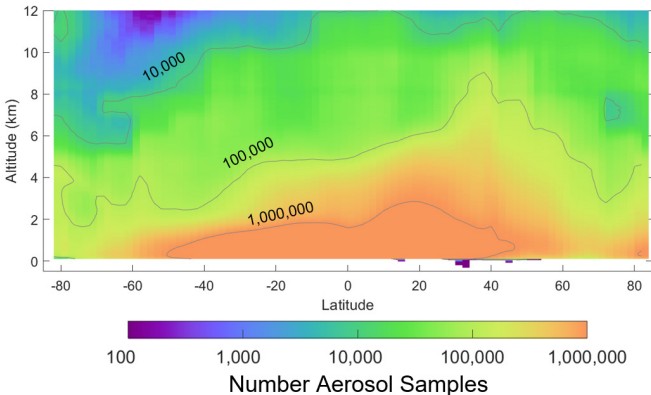

5    **Figure 7. Zonal total number of aerosol samples reported by the level 3 product prior to quality screening for 2007–2016 at night, all-sky.**

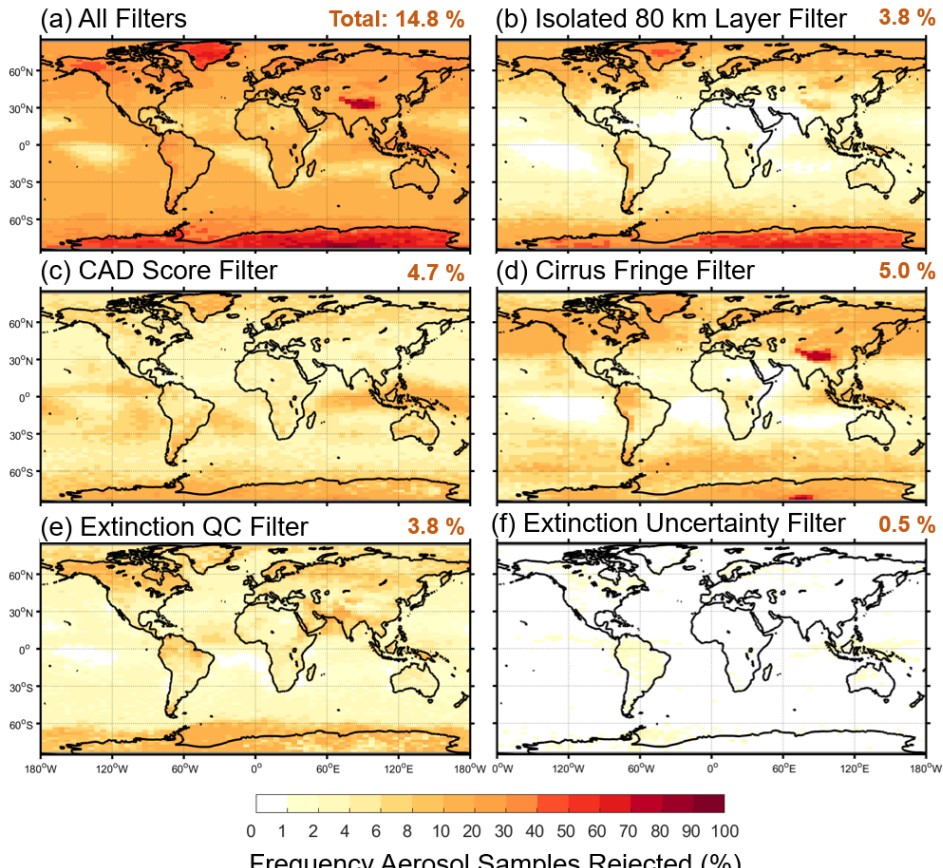

**Figure 8. Frequency of level 3 aerosol samples rejected by the indicated filter out of all aerosol detected as reported by the level 3 product for 2007–2016 at night, all-sky. Global total rejection frequencies are indicated in the panel titles.**

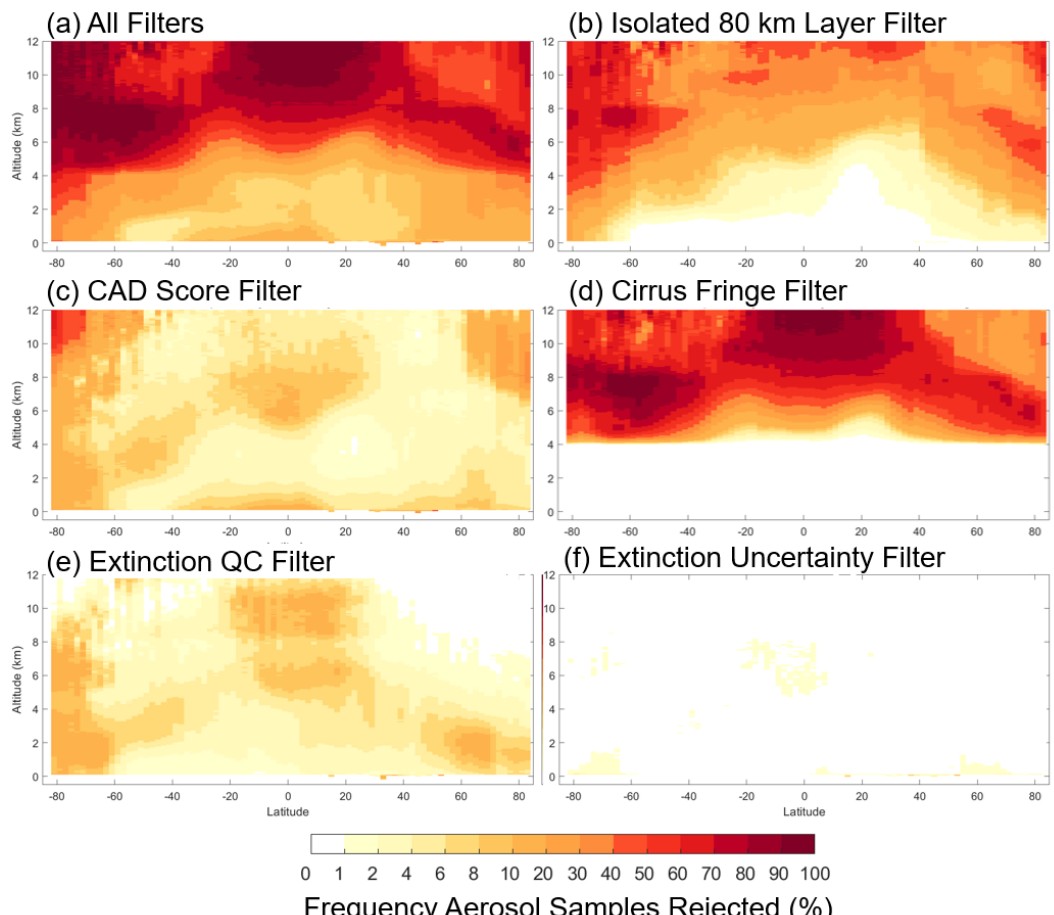

**Figure 9. Zonal frequency of aerosol samples rejected by the indicated filter out of all aerosol detected as reported for by the level 3 product for 2007–2016 at night, all-sky.**

## 5.1 Isolated 80 km aerosol layer filter

*Level 2 aerosol layers detected at 80 km horizontal resolution that are not in contact with another aerosol layer are assumed to be noise-induced misclassifications and are rejected.*

      In low SNR regions such as beneath optically dense clouds, some detected features may actually be artifacts due to noise rather than legitimate aerosol. These noise artifacts are usually detected at 80 km horizontal averaging resolution. This filter reduces the occurrence of noise misclassified as aerosol.

      In scenes with significant overlying attenuation, features may be detected at 80 km resolution after more strongly scattering features have been detected and removed. However, if these layers are isolated and not in contact with other aerosol layers, it is possible they represent detection artifacts rather than actual aerosol layers. These weakly scattering layers contribute little to monthly mean AOD, but would affect the spatial distribution of level 3 aerosol occurrence if accepted. For this reason, the level 3 algorithm rejects isolated aerosol layers detected at 80 km resolution.

For the 10-year evaluation period, the isolated 80 km aerosol layer filter rejected 3.8 % (5.9 %) of samples at night (day). Daytime rejection is higher because solar noise reduces SNR, making coarser averaging necessary for layer detection relative to night. The largest frequency of rejection is over the poles and over Greenland (Fig. 8(b)). Aerosol is most often rejected at altitudes where deep convective clouds are expected (Mace and Wrenn, 2013): above 8 km at the equator and lower towards the poles (Fig. 9(b)). During the day, larger rejection frequencies occur at lower altitudes: ~6 km over the equator and near 4 km towards the poles (Fig. S4(b)). In this case, legitimate aerosol may be rejected because weakly scattering aerosol layers are not always detected due to the reduced SNR. Therefore, it becomes less likely for an aerosol layer detected at 80 km resolution to be in contact with another, and the possibility of rejection is higher. This phenomenon is exacerbated by high albedo surfaces, which induce noise through the profile, limiting the fidelity of feature detection.

## 5.2 Filters for clouds misclassified as aerosol

Another source of error that can bias level 3 aerosol statistics is clouds misclassified as aerosol. Two filters are employed to reject layers suspected of being misclassified clouds. The first filter uses the CAD score, a built-in level 2 quality flag with a strong empirical foundation. The second filter uses a spatial proximity test to reduce the impact of the tenuous edges of cirrus clouds that are misclassified as aerosol.

### 5.2.1 CAD score filter

*Level 2 aerosol layers with CAD score outside the range [−100, −20] are rejected because there is no confidence in cloud-aerosol discrimination.*

The cloud-aerosol discrimination (CAD) algorithm evaluates five CALIOP observables to classify layers as aerosol or cloud: 532 nm layer-mean attenuated backscatter ($< \beta'_{532} >$), layer-mean attenuated color ratio ($\chi' = < \beta'_{1064} > / < \beta'_{532} >$), layer-integrated volume depolarization ratio ($\delta_v$), latitude, and altitude. These five observables are evaluated against five dimensional probability density functions of identical observables where aerosol and cloud layers have been manually classified (Liu et al., 2009). For the idealized case, aerosol layers tend to have lower values of $< \beta'_{532} >$ and $\chi'$ compared to clouds, and aerosol layers exist most often at lower altitudes. There is often overlap between the cloud and aerosol probability distributions, so type classification confidence is reduced for layers having measured values within the overlap region.

In order to quantify the classification confidence, a CAD score ranging between −100 and 100 is computed for each layer (Liu et al., 2009). A CAD score of −100 indicates that the feature is very likely an aerosol layer, and a CAD score of +100 indicates that the feature is very likely a cloud. There is no confidence in cloud-aerosol discrimination for features with |CAD score| < 20. For the year 2010 at night in version 3, over 85 % of aerosol layers have CAD score < −90 and around 4 % have CAD score > −20. The remaining 11 % have intermediate levels of confidence.

Aerosol layers having CAD scores outside the range of [−100, −20] are rejected because there is no confidence in discriminating aerosol from cloud. These layers tend to have larger overlying attenuation relative to those with CAD score < −20, which reduces the SNR of the measurements and degrades the fidelity of CAD classification (Fig. 10). No-confidence CAD scores also indicate a high probability of layer detection artifacts where noise spikes cause the feature finder to detect layers that do not actually exist.

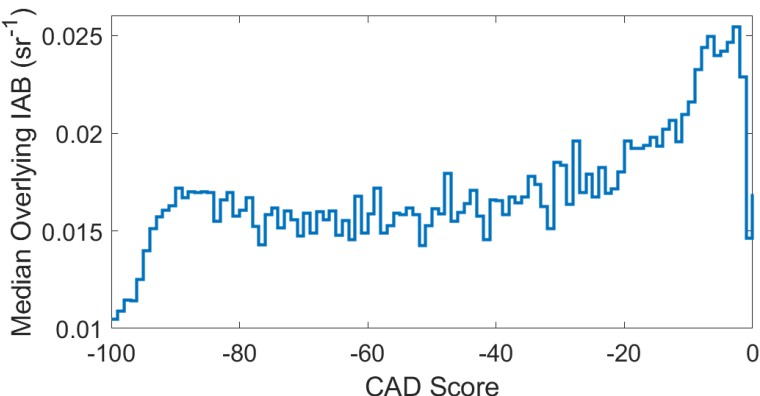

**Figure 10. Median overlying integrated attenuated backscatter (IAB) for aerosol layers having the indicated CAD score for 2010 at night, global.**

Note that filtering with a very restrictive CAD score range can significantly alter the $\bar{\bar{\sigma}}$ profile. In Fig. 11, the restrictive CAD score ranges of [−100, −90] and [−100, −99] significantly reduce $\bar{\bar{\sigma}}$ relative to the [−100, −20] range. The CAD algorithm finds weakly scattering features to be more aerosol-like and receive higher confidence CAD scores relative to strongly scattering features, which appear more cloud-like, lowering the CAD score. Thus, higher confidence aerosol CAD scores tend to be associated with lower $\sigma$ values, which alters the $\bar{\bar{\sigma}}$ profile shape. Rejecting layers with CAD scores outside the [−100, −20] range removes low confidence layers with minimal impacts on AOD (Sect. 6.1).

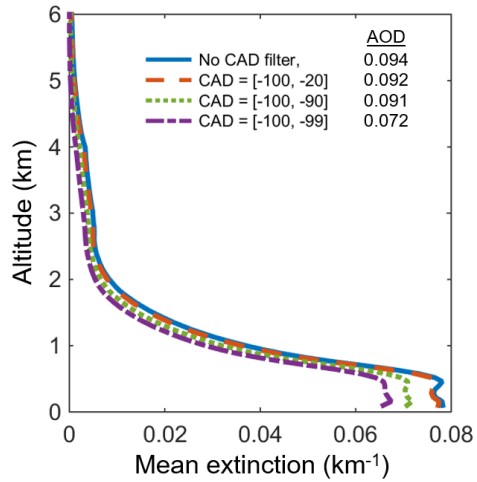

**Figure 11. Mean extinction without the CAD score filter (blue solid line) and with three different CAD score ranges (dashed lines) for 2010 at night, all-sky, ocean-only, 50° S–50° N.**

For the 10-year evaluation period, the CAD score filter rejected 4.7 % (5.1 %) of samples at night (day). Most rejection occurs over Antarctica, Greenland, and in the tropics (Fig. 8(c)). At the poles, ice clouds can be misclassified as dust due to enhanced $\delta_v$, increasing the rejection frequency of no-confidence CAD scores. Though the rejection frequency in the polar regions is high, the total number of aerosol samples is low (Fig. 6). Rejection frequencies are elevated due to signal attenuation along the lower portions of deep convection in the tropics and along frontal systems at higher latitudes; above 4 km at the equator and at progressively lower altitudes poleward (Fig. 9(c)). Rejection frequencies are also elevated below 1 km along the tropics where zero confidence CAD scores exist for some surface-attached layers.

### 5.2.2 Misclassified cirrus fringe filter

*Level 2 aerosol layers above 4 km that are in contact with ice clouds are rejected as misclassified cirrus fringes.*

At times, the tenuous edges of cirrus (i.e., cirrus fringes) are misclassified as aerosol. A prime example is shown in Fig. 12 where "aerosol" is detected along the edges and beneath an extensive cirrus layer. These misclassifications commonly occur in regions of extensive cirrus and complex cloud layering. They occur most often at night where higher SNR allows more frequent detection of optically thin layers after averaging to 20 km and 80 km horizontal resolutions.

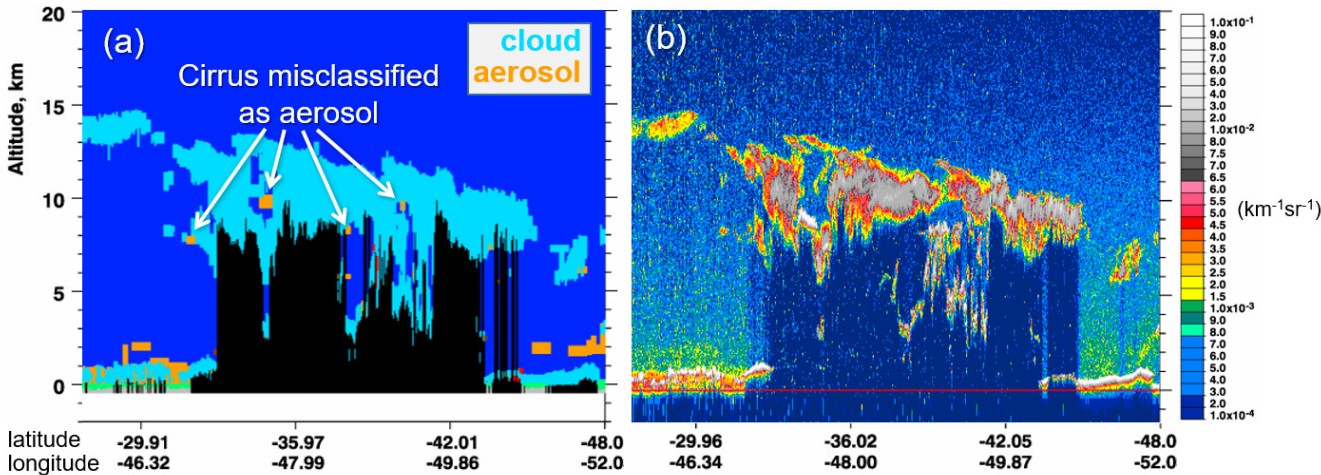

**Figure 12. (a) Feature type classification and (b) total attenuated backscatter showing cirrus misclassified as aerosol for the version 3 granule 2011-11-10T03-54-52ZN.**

Even though these layers are optically thin, the frequency of aerosol detection at these altitudes is low and even a few misclassified cirrus fringes can skew the representativeness of aerosol presence. For instance, Fig. 13 shows the vertical profile of dust detection frequency in the southern Pacific Ocean, where high-altitude dust is not expected. The aerosol classified as dust within the marine boundary layer (albeit infrequently, < 0.3 %) is likely associated with residual cloud

layers detected at 1/3 km resolution affecting $\delta_v$, causing aerosol subtyping misclassifications. However, the enhanced frequency of dust detection at higher altitudes is the main issue addressed by this filter: when the cirrus fringe filter is not applied (blue profile), the peak altitude of dust frequency appears at nearly 7 km. Since there is little evidence to support dust at these altitudes in this region, dust frequency appears overestimated (again, infrequently at ~0.3 % or less).

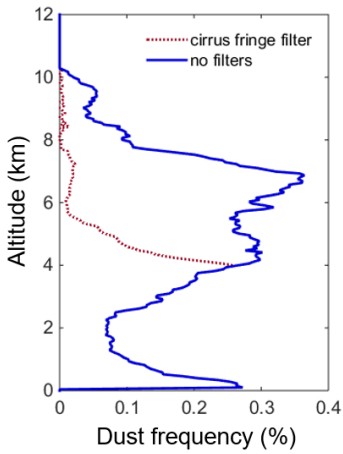

**Figure 13. Dust detection frequency ($100 \times N_{dust} / N_{all\ aerosol}$) with and without the cirrus fringe filter for September – November 2010 at night over the south Pacific Ocean [30° S, 55° S; 80° W, 180° W].**

Two phenomena are at work here. First, clouds transition into cloud-free environments continuously, becoming optically thinner with further distance from cloud (Koren et al., 2007). When small amounts of cloud particles are included in a 20 km or 80 km horizontal resolution average, both $< \beta'_{532} >$ and $\chi'$ are reduced as the molecular scattering contribution begins to dominate. Small $< \beta'_{532} >$ and low $\chi'$ resembles aerosol to the CAD algorithm, hence they are classified as such, often with high-confidence CAD scores. The presence of ice elevates $\delta_v$, causing many of these layers to be classified as dust. The second phenomenon is overlying attenuation, which can cause features detected beneath cirrus clouds to be misclassified as aerosol. These layers also can have high-confidence aerosol CAD scores that cannot be removed by the CAD score filter alone. For the purposes of level 3, these layers are considered misclassified cirrus fringes.

There are of course legitimate reasons that aerosol could exist adjacent to cirrus and other types of ice clouds (e.g., pyrocumulonimbus (Fromm et al., 2010)). Deep convection can loft aerosols to high altitudes where they become ice nuclei for cirrus or remain in an unfrozen state (Froyd et al., 2010; Chakraborty et al., 2015). The "Asian Tropopause Aerosol Layer" is hypothesized to loft pollution during the Asian summer monsoon (Vernier et al., 2011, 2015). Dust storms can loft dust, particularly effective ice condensation nuclei, to high enough altitudes to co-exist with ice clouds (Klein et al., 2010). Volcanic aerosol injected to high altitudes can also act to seed cirrus clouds (Campbell et al., 2012b). For CALIOP, however, misclassification is the most likely explanation in most cases where isolated aerosol layers are found in direct contact with spatially extensive cirrus layers, and not the sudden appearance of previously undetected aerosol.

Therefore, to exclude misclassified cirrus fringes, "aerosol" layers are rejected when their bases are above 4 km and they are adjacent to ice clouds; i.e., clouds classified as either randomly or horizontally oriented ice by the CALIOP ice/water phase retrieval (Hu et al., 2009) and having a cloud top temperature less than 0° C. The 4 km altitude threshold limits the magnitude of error that would be made by rejecting legitimate aerosol in the lower troposphere where aerosol and clouds are more likely to coexist. For example, 95 % of all aerosol layers detected in 2010 are below 4 km (global). Meanwhile, 11 % of all ice clouds are also detected below this altitude. Ice clouds below 4 km are even more frequent at high latitudes: comprising ~22 % of all ice clouds at latitudes higher than 50° N/S in 2010. The global 4 km threshold thereby protects the majority of legitimate aerosols from being incorrectly rejected, albeit with the possibility of some remaining cirrus fringes at high latitudes.

While dust detected by CALIOP is typically at or below altitudes where ice clouds are found, one region where dust and ice clouds are expected to coexist is east of Asia during northern hemisphere spring. Dust from the Taklimakan and Gobi deserts are frequently lofted to high altitudes and transported across the Pacific Ocean (Yu et al., 2012). As a check on whether these legitimate dust layers adjacent to cirrus are being erroneously rejected by the filter, Figure 14 shows that dust $\bar{\sigma}$ above 4 km is still well represented after the cirrus fringe filter is applied since most dust plumes are not in contact with ice clouds. The reduction in full column dust AOD is small in this case, about 7 %. On the other hand, dust frequency is reduced substantially above 4 km in the southern Pacific Ocean where dust is not expected (Fig. 13, red line), preventing these misclassified fringes from contributing to $\bar{\sigma}$.

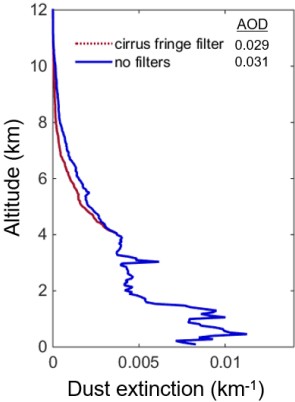

**Figure 14. Mean dust extinction with and without cirrus fringe filter for March – May 2010 at night over the Asian dust outflow region [30° N, 60° N; 140° E, 180° E].**

For the 10-year evaluation period, the cirrus fringe filter rejected 5.0 % (1.3 %) of all aerosol layers at night (day). Nighttime rejection frequencies of 10–20 % occur poleward of 30° in both hemispheres and over the Asian maritime continent (Fig. 8(d)). Daytime rejection frequencies are lower in these regions, typically less than 5–10 % (Fig. S3(d)). The highest relative rejection frequencies over the Tibetan Plateau, Antarctica, and Greenland are associated with very low

aerosol detection rates (Fig. 6). Rejection rates correlate with the frequency of cirrus, with nighttime rejection rates > 90 % above 10 km at the equator.

## 5.3 Filters that remove extinction retrieval issues

Two metrics reported in the level 2 aerosol profile product are used to assess the quality of extinction retrievals: the extinction QC flag and the extinction uncertainty. The extinction QC flag summarizes the final state of the extinction retrieval solution, while the extinction uncertainty provides an estimate of systematic and random errors. Note that these filters do not remove negative extinction values. Though unphysical, negative extinction values can result from signal noise and must be retained to prevent biasing $\bar{\sigma}$ high.

### 5.3.1 Extinction QC filter

*Level 2 aerosol layers with extinction QC flags not equal to 0, 1, 16, or 18 are rejected as low-confidence extinction retrievals.*

Generating an extinction solution requires a lidar ratio ($S_p$) estimate appropriate for the layers being solved. If $S_p$ is not appropriate, it must sometimes be adjusted to guarantee convergence throughout the entire profile. A level 2 extinction QC flag (extQC) summarizes the final status of the extinction solution for each layer, indicating solutions for which the initial $S_p$ was unchanged, adjusted, or derived directly from measurements (Table 2). Layers exhibiting any of the special error states in Table 2 are rejected because they indicate convergence could not be achieved or internal quality control checks have trapped spurious solutions.

Table 2. Extinction QC flag values, definitions, and frequencies out of all aerosol layers for 2007–2010, night & day.

| Extinction QC flag values and definitions | Frequency (%) |
|---|---|
| 0  – Lidar ratio is default value, unchanged | 96.3 |
| 1  – Lidar ratio is measured | 0.01 |
| 2  – Lidar ratio is reduced from default value | 1.73 |
| 16 – Lidar ratio is default value, layer is opaque | 1.44 |
| 18 – Lidar ratio is reduced from default value, layer is opaque | 0.31 |
| 4, 8, 32, 64, 128, 256 – special error states | 0.19 |

Layers with extQC = 0 occur most frequently (> 95 % of all retrievals). This value indicates that the layer was solved with the default $S_p$ for the layer subtype, without adjustment during the retrieval process. Note however, this does not guarantee that the extinction solution accurately describes the atmospheric conditions. It just means that the retrieval

converged within specified limits at all analyzed range bins while using the default $S_p$. For an individual aerosol layer, the uncertainty of a successful extQC = 0 aerosol extinction retrieval is at least 30–50 % based on estimates of the natural variability of $S_p$ for each aerosol subtype (Omar et al., 2009).

Layers with extQC = 1, 16, and 18 are also accepted. Instead of a default $S_p$, layers with extQC = 1 derive an optimal value of $S_p$ from measurements of layer two-way transmittance, thereby reducing systematic uncertainty due to $S_p$ selection (Young and Vaughan, 2009). These are the least frequent of all solutions for aerosol layers, however (~0.01 % of all retrievals). A value of extQC = 16 indicates opaque layers where, like extQC = 0, the default $S_p$ is unchanged during the retrieval. These layers are optically thick and can contribute substantially to $\bar{\sigma}$. Similarly, extQC = 18 indicates opaque layers, but the initial $S_p$ is reduced during the retrieval process. The initial $S_p$ is also reduced for layers with extQC = 2, but these layers are transparent.

$S_p$ is reduced for layers having extQC = 2 or 18 because the initial values are too large to permit a solution. This can either occur due to incorrect aerosol subtype selection or because there is a large difference between the default $S_p$ and true value due to natural variability. It can also occur when the optical depth retrieved for overlying layers is overestimated, resulting in over-corrected attenuated backscatter coefficients within the layer being solved (Young and Vaughan, 2009). As the layer optical depth increases, the retrieval becomes increasingly sensitive to errors in lidar ratio selection (Young et al., 2013). For opaque layers, the retrieval becomes especially sensitive, causing the extQC = 18 condition to occur for even small errors in lidar ratio selection. Due to natural variability of aerosol lidar ratio, even an unbiased initial value would be expected to cause extQC = 18 about half the time. For transparent layers, on the other hand, (typically having AOD $\ll$ 1), the extQC = 2 condition only arises from large errors in the initial $S_p$ selection or from errors incurred while correcting for overlying attenuation. This can be problematic because the retrieval algorithm only reduces $S_p$ sufficiently to permit a successful retrieval – yet, the final $S_p$ might still be too large. The result tends to be a significant high bias in retrieved extinction in the version 3 level 2 algorithm for aerosol layers with extQC = 2. For these reasons, layers having extQC = 18 are accepted whereas those with extQC = 2 are rejected to avoid potential high-biases in level 3 $\bar{\sigma}$.

Note that all aerosol extinction coefficients *below* layers rejected by the extinction QC filter should also be rejected because their solutions are affected by the low-confidence transmittance estimates from overlying rejected layers. Even though this was not done in the version 3 level 3 aerosol product, future versions will adopt this convention.

The extinction QC filter is particularly active in regions where it is plausible to expect aerosol $S_p$ reductions. Fig. 15(a) shows high rejection frequencies over the Arabian Sea for the 10-year evaluation period, with rejection frequencies approaching 40% in the June – August (JJA) season (Fig. S5(a)). In this region, dust ( $S_p \equiv 40$ sr) commonly mixes with marine aerosol ( $S_p \equiv 20$ sr) and this mixture is misclassified as polluted dust by the version 3 aerosol typing algorithm (the

triple bars denote that these are default assigned values). Classification as polluted dust ($S_p \equiv 55$ sr) significantly overestimates the lidar ratio of a dust/marine mixture, which would fall in the range 20 sr $< S_p <$ 40 sr (Omar et al., 2018), causing the need to reduce $S_p$. A similar argument can be made for the high rejection frequency of Saharan dust samples over the central Atlantic Ocean. Rejections over the Antarctic, however, are more often caused by special error states listed

in Table 2 rather than the need to adjust $S_p$.

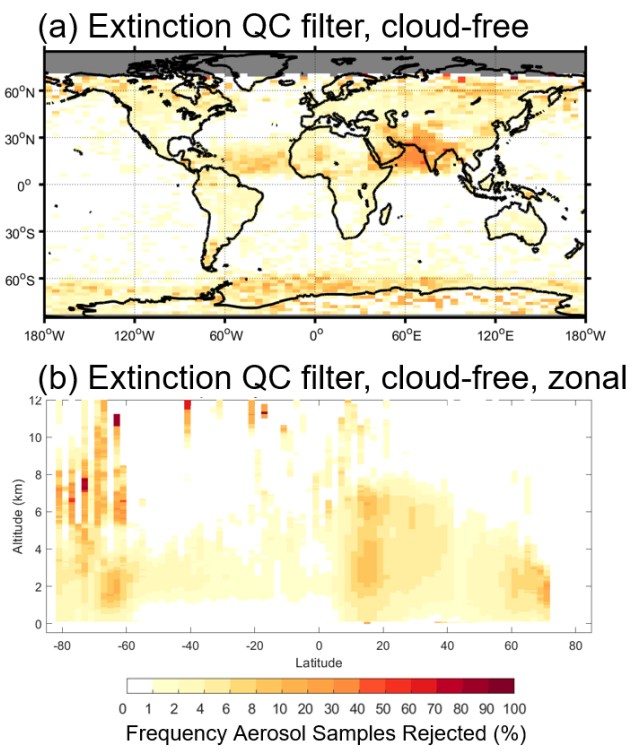

**Figure 15. (a) Column and (b) zonal frequency of aerosol samples rejected by the extinction QC filter out of all aerosol detected for 2007–2016 at night, cloud-free.**

For the all-sky 10-year evaluation period, the extinction QC filter rejected 3.8 % (3.0 %) of samples at night (day).

The rejection rate for cloud-free is half that, about 2 % night and day. This is expected due to errors incurred while solving overlying cloud layers. The locations of the highest all-sky rejection frequencies are similar to those of cloud-free (cf. Figs. 8(e) and 15(a)), but with an additional 2–6 % rejected over the oceans and an overall increase in rejections due to retrieval errors caused by cloud cover. For the cloud-free sky condition, aerosol sample rejection is confined to altitudes below 6–8 km in most regions (Fig. 15(b)). Zonal rejection frequency is 4–6 % below 4 km at latitudes between 40° N and 60° N,

corresponding to land-based aerosol sources in the December – February (DJF) season (Fig. S5(b)). Within the Saharan dust belt and over the Arabian Sea, zonal rejection frequencies of ~8 % occur between 1 and 6 km in altitude in the JJA season. All-sky zonal rejection frequency is higher for these regions, approaching 10–20 % (Fig. 9(e)). Aerosol samples are also

rejected above 8 km over the tropics in the all-sky condition, approaching similar rejection frequencies due to overlying cloud cover.

### 5.3.2 Extinction uncertainty filter

*Level 2 aerosol extinction samples having extinction uncertainty equal to 99.99 km$^{-1}$ are rejected. Aerosol extinction coefficients in all range bins directly below these samples are also rejected since their extinction solutions are affected.*

Extinction uncertainty ($\Delta\sigma$) reported in the level 2 profile products, provides an estimate of random and systematic errors at each range bin (Young et al., 2013, 2016). Uncertainty accumulates during the top-down retrieval and propagates to solutions at lower altitudes. Aerosol layers near the surface therefore tend to have larger $\Delta\sigma$ compared to

10 those at higher altitudes because there are more likely to be overlying layers. In the level 2 data product, $\Delta\sigma$ is limited to a maximum value of 99.99 km$^{-1}$. This extreme value usually occurs where the retrieved extinction is increasing rapidly due to the use of $S_p$ values that are too large or from significant renormalization errors beneath higher layers (Young et al., 2013). As shown in Fig. 16(a), uncertainties of 99.99 km$^{-1}$ are often associated with very large aerosol extinction values. These large, highly uncertain extinction values will bias the level 3 average high if not rejected.

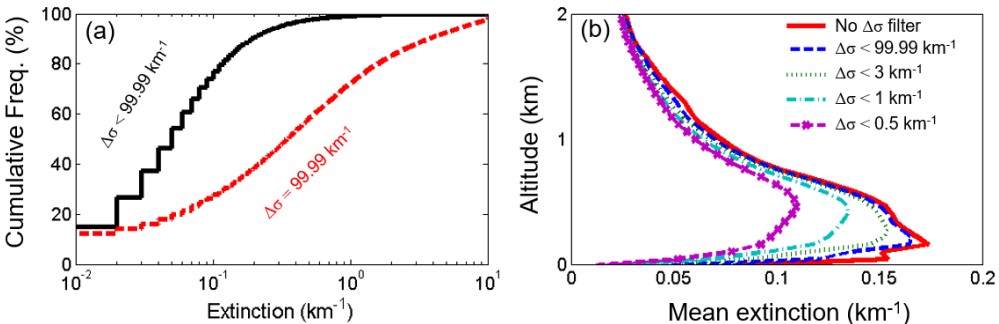

**Figure 16. (a) Cumulative frequency distributions of level 2 $\sigma$ where $\Delta\sigma \neq$ 99.99 km$^{-1}$ (black) and $\Delta\sigma$ = 99.99 km$^{-1}$ (red) for August 2007; (b) level 3 $\bar{\sigma}$ profiles without the $\Delta\sigma$ filter (red) and with varying upper limits on the $\Delta\sigma$ filter threshold (dashed lines) for mid-Atlantic Ocean, 2007, all-sky at night.**

In the level 3 product, only retrievals with $\Delta\sigma$ = 99.99 km$^{-1}$ are rejected. Lower threshold values can make the

20 filter extremely aggressive, as seen in Figure 16(b), which shows the impact of four $\Delta\sigma$ thresholds on $\bar{\sigma}$. Since $\sigma$ tends to be largest near the surface, the larger $\sigma$ values are preferentially rejected and the $\bar{\sigma}$ profile shape changes to a stronger degree for subsequently lower $\Delta\sigma$ thresholds. However, the $\Delta\sigma$ = 99.99 km$^{-1}$ threshold affects the $\bar{\sigma}$ profile shape by the least while still rejecting solutions that are untrustworthy.

For the 10-year evaluation period, the extinction uncertainty filter rejects a small number of aerosol samples: 0.5 %

(0.7 %) at night (day). Rejections tend to occur more frequently over land near the surface and within the intertropical

convergence zone, though not markedly so (Figs. 8(f) and 9(f)). At night, around 1.5 % of samples are rejected above 4 km within the tropics (Fig. 9(f)), whereas during the day 3–4 % of samples are rejected in this region (Fig. S4(f)). Even though rejection frequencies are low, the impact on the $\bar{\sigma}$ profile can be significant near the surface. Figure 17 shows regional $\bar{\sigma}$ for northeast South America with and without the $\Delta\sigma$ filter. Spuriously large, highly uncertain extinction values just above the

5 surface, which distort the near-surface profile shape, are rejected; while having only a small impact on global mean AOD (a reduction of ~10 %).

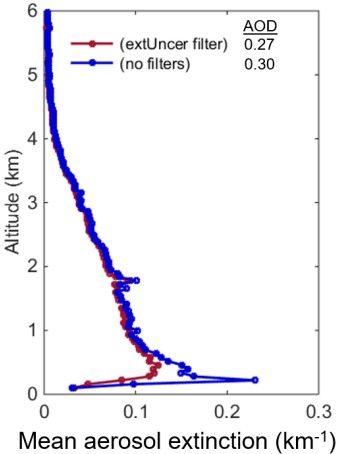

**Figure 17. Level 3 $\bar{\sigma}$ with the extinction uncertainty filter (red) and with no filters (blue) over the South America region (Table A1) for 2010 at night, all-sky.**

**5.4 Negative signal anomaly mitigation**

*All level 2 atmospheric samples (aerosol, cloud, and clear-air) are ignored within 60 m of the local surface to avoid aerosol extinction affected by the negative signal anomaly.*

The final quality filter addresses an intermittent phenomenon referred to as the "negative signal anomaly" (NSA). This signal artifact occurs when the level 1B attenuated backscatter becomes strongly negative preceding a strongly

scattering target such as the surface. The NSA is intermittent, but tends to occur in sequences of adjacent profiles within latitude bands that vary seasonally. If these negative spikes are treated as part of a surface-attached aerosol layer, they can produce large negative aerosol extinction values just above the surface. An example of the NSA is evident in the attenuated backscatter signal shown in Fig. 18(a). The retrieved $\sigma$ value from this data can be strongly negative, or worse, it can bias the signal low and yet still remain positive. Figure 18(b) shows three aerosol extinction profiles retrieved from the attenuated

backscatter in Fig 18(a) along separate 5 km segments containing the NSA. While the strongly negative values adjacent to the surface are readily apparent for the extinction profiles in *this* example, positive $\sigma$ values that are biased low are not as easy to detect. This can occur because $\sigma$ is retrieved after averaging 15 level 1B attenuated backscatter profiles to 5 km

horizontal resolution. If only some of the level 1B profiles are affected by the NSA, the average backscatter can still be positive, yet biased low.

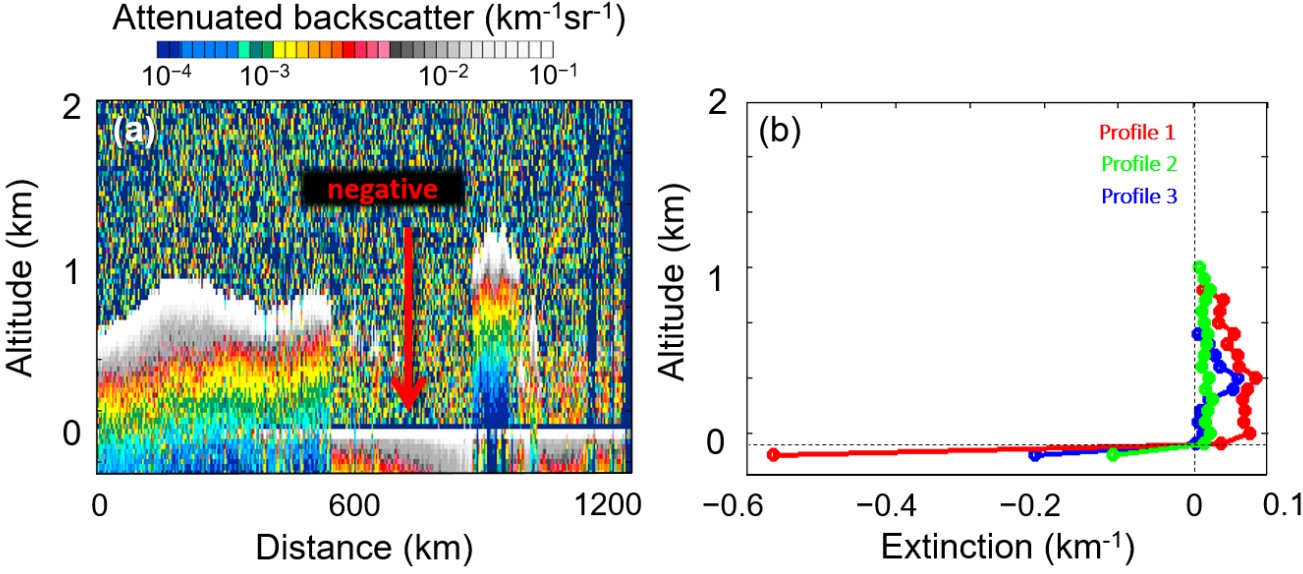

Figure 18. Example of NSA in the granule 2006-07-27T00-22-12ZN, centered at ~25° S, 10.5° E. (a) Level 1B total attenuated backscatter. (b) Level 2 aerosol extinction profiles.

In order to prevent near-surface $\sigma$ affected by the NSA from biasing $\bar{\sigma}$, all atmospheric samples within 60 m of the local surface are ignored. This approach was adopted because it is difficult to know when the NSA has influenced $\sigma$ at the surface. An example of the impact of this NSA mitigation is shown in Fig. 19. AOD increases by roughly 5–10 % in level 3 profiles affected by the NSA (based on values > 1 within the red boxes) because strongly negative near-surface $\sigma$ is rejected. Conversely, the NSA in this example is also present along the equator, and yet excluding these $\sigma$ values does not increase AOD, illustrating the difficulty of predicting the influence of the NSA on retrieved extinction. AOD also decreases by roughly 5 % on average in unaffected regions, a consequence of this conservative strategy. Note that recently released version 4 level 1B and level 2 data products have mitigation procedures in place to remove the effect of the NSA on $\sigma$ by excluding affected level 1B backscatter (Vaughan et al., 2018). Future versions of the level 3 aerosol product using version 4 data should no longer require the mitigation strategy described here.

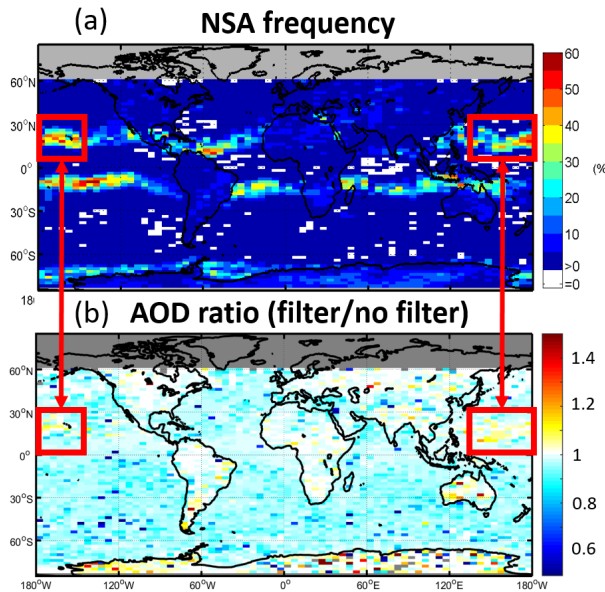

**Figure 19. (a) Frequency of level 1B profiles containing the negative signal anomaly (NSA) in parallel 532 nm lidar channel and (b) ratio of level 3 mean AOD with and without NSA mitigation filter for July 2008 at night. Note that the geographic location of enhanced NSA frequency changes seasonally.**

## 6 Impact of quality screening on mean extinction and AOD

The final section of this paper quantifies the impact of quality screening on level 3 $\bar{\bar{\sigma}}$ and AOD. First, the relative impact of individual quality filters are compared, followed by an assessment of the overall impact when all quality filters are applied. Mathematical definitions of metrics used to quantify changes in $\bar{\bar{\sigma}}$ and AOD are given in Appendix B whereas their interpretations are described below.

### 6.1 Individual filters

The quality filters implemented in the level 3 aerosol algorithm influence $\bar{\bar{\sigma}}$ profiles and AOD to varying degrees. This section quantitatively compares the influence of the quality filters on $\bar{\bar{\sigma}}$ and AOD when applied independently in order to identify the most influential filters. Though the rejection frequency of different filters varies regionally and seasonally, a globally averaged annual summary is sufficient to establish their relative ranking.

Figure 20 summarizes the impact of quality filters on the global $\bar{\bar{\sigma}}$ profile, the frequency of rejection, and filter aggressiveness. The aggressiveness metric $Agr(z)$ indicates the effectiveness of sample rejection on changing $\bar{\bar{\sigma}}$, with larger values indicating the filter is more aggressive at changing $\bar{\bar{\sigma}}$ than other filters. It is computed as the change in $\bar{\bar{\sigma}}$ per reduction in number of aerosol samples due to filtering (Eq. (B1)). For context, Fig. 20(b) shows the number of unfiltered aerosol samples, which decreases with increasing altitude.

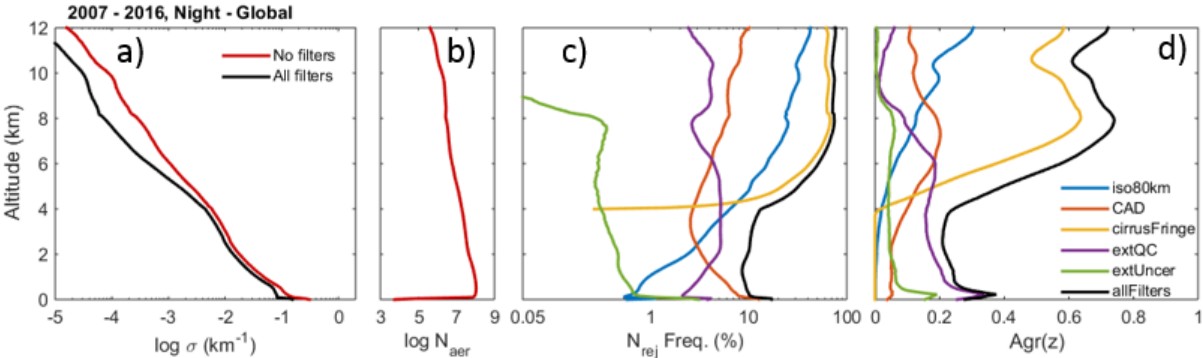

**Figure 20. (a) Mean extinction with and without quality filters, (b) number of unfiltered aerosol samples, (c) frequency of aerosol samples rejected, and (d) filter aggressiveness (Eq. (B1)) smoothed vertically over 600 m for 2007–2016 at night, all-sky.**

The filters rejecting the highest frequency of samples below 2 km are the CAD and extinction QC filters (Fig. 20(c)). Above 4 km, the cirrus fringe and isolated 80 km layer filters dominate the aerosol sample rejection. Based on the $Agr(z)$ metric in Fig. 20(d), the extinction QC filter is the most aggressive in changing $\bar{\sigma}$ at low altitudes (< 4 km) despite the CAD filter rejecting a higher frequency of samples below 2 km. This demonstrates that $\sigma$ rejected by the extinction QC filter is often quite large relative to $\sigma$ rejected by the CAD filter. Above ~6 km, the cirrus fringe filter is by far the most aggressive at changing $\bar{\sigma}$. A similar conclusion is expected for small (~ 1 km) perturbations of the 4 km altitude threshold for this filter. Daytime $\bar{\sigma}$ at these high altitudes is influenced by both the cirrus fringe and isolated 80 km layer filters to a similar degree (Fig. S6(d)).

As a global summary of the impacts of quality filtering on $\bar{\sigma}$ and AOD, Table 3 compares four metrics for each filter applied independently: quality filtered AOD, percent change in AOD (ΔAOD), change in extinction scale height ($\Delta z_{63}$, Eq. (B3)), and sample-weighted mean filter aggressiveness (**Agr**, Eq. (B3)). While mean AOD and its percent change ΔAOD characterize the full-column impact on strongly scattering aerosol, $\Delta z_{63}$ is an indicator of impacts on the vertical distribution. Positive values indicate that the altitude containing 63 % of total AOD has moved upward after quality filtering.

The most influential quality filters are the extinction QC and extinction uncertainty filters, respectively. The extinction QC filter is responsible for the largest reductions in AOD: −19 % (global ocean) to −28 % (global land). This accounts for all but 3–5 % of total AOD reductions due to all filters together. The extinction uncertainty filter is the second-most impactful filter in terms of AOD reduction, but with reductions 2–3 times smaller than the extinction QC filter. These same two filters are also responsible for increasing $\Delta z_{63}$ over land by 180–240 m (with the extinction QC filter causing the larger increase). The altitude containing the bulk of AOD increases because these filters are more aggressive in the lowest altitudes (Fig. 20(d)), leaving higher-altitude aerosol to contribute more to the total AOD. For the remaining filters, $\Delta z_{63}$ is zero or decreases by 60 m (over land and ocean) because these filters act upon layers at higher altitudes. Reducing $\bar{\sigma}$ at

higher altitudes allows lower-altitude aerosol to contribute more to total AOD. In terms of filter aggressiveness **Agr**, the extinction QC filter is the most aggressive of all filters with the cirrus fringe filter being the second most aggressive, albeit at higher altitudes (Fig. 20(d)).

Table 3. Global metrics comparing changes in level 3 mean AOD when all filters are applied (top row) and when each filter is applied independently (remaining rows) for global ocean and global land for 2007–2016 at night, all-sky: AOD with all filters, $\Delta$AOD = percent change in AOD, $\Delta z_{63}$ = difference in 63 % extinction scale heights (all filters – no filters; Eq. (B3)), **Agr** = aerosol sample-weighted mean of filter extinction impact profile (Eq. (B4)). Samples at altitudes ≤ 0.039 km are excluded due to low sample counts.

| | Global ocean | | | | Global land | | | |
|---|---|---|---|---|---|---|---|---|
| | AOD | $\Delta$AOD (%) | $\Delta z_{63}$ (m) | **Agr** | AOD | $\Delta$AOD (%) | $\Delta z_{63}$ (m) | **Agr** |
| All filters | 0.09 | −24 | 0 | 0.38 | 0.21 | −31 | 240 | 0.41 |
| Isolated 80 km | 0.11 | −0.5 | −60 | 0.02 | 0.31 | −0.4 | 0 | 0.02 |
| CAD | 0.11 | −7 | −60 | 0.08 | 0.30 | −3 | 0 | 0.05 |
| Cirrus fringe | 0.11 | −1 | −60 | 0.10 | 0.31 | −1 | 0 | 0.12 |
| Extinction QC | 0.09 | −19 | 0 | 0.17 | 0.22 | −28 | 240 | 0.20 |
| Extinction uncertainty | 0.11 | −7 | 0 | 0.06 | 0.27 | −13 | 180 | 0.06 |

## 6.2 Net impacts

This section examines changes in $\bar{\sigma}$ and AOD due to quality filtering in twelve key regions (Fig. 21). These regions are roughly the same as those defined by Yu et al. (2010) and adapted by Koffi et al. (2016) to characterize dust, marine, biomass burning, and industrial aerosols for global aerosol model comparisons with CALIOP retrievals.

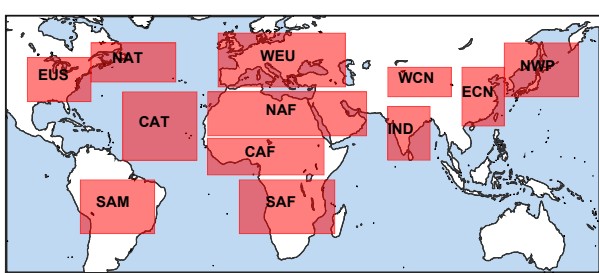

Figure 21. Region definitions, similar to those defined by Yu et al. (2010): EUS = Eastern United States, WEU = Western Europe, IND = India, ECN = Eastern China, NAT = North Atlantic Ocean, CAT = Central Atlantic Ocean, NWP = Northwest Pacific Ocean, NAF = North Africa, WCN = Western China, SAM = South America, CAF = Central Africa, SAF = Southern Africa. Geographic boundaries are specified in Table A1.

Regional $\bar{\sigma}$ profiles are shown in Fig. 22 before filtering and after applying all quality filters. Median surface elevations are shown to indicate altitudes where the number of samples averaged begins to decrease (often rapidly) relative to higher altitudes, thereby increasing the uncertainty in the mean values being compared. Impacts of applying just the extinction QC filter are shown as the dashed green line. The $\bar{\sigma}$ profile differences are very slight between the extinction QC filter-only and all-filters cases. Most differences occur at the surface where the extinction uncertainty filter has rejected suspiciously large extinction values; e.g., central Africa (CAF) region. In order to compare changes in $\bar{\sigma}$ and AOD quantitatively within these regions, Fig. 23 presents the same change metrics defined in the Sect. 6.1. Numerical values for these metrics, their seasonal counterparts, and AODs are tabulated in Table S1 and shown in Fig. S7.

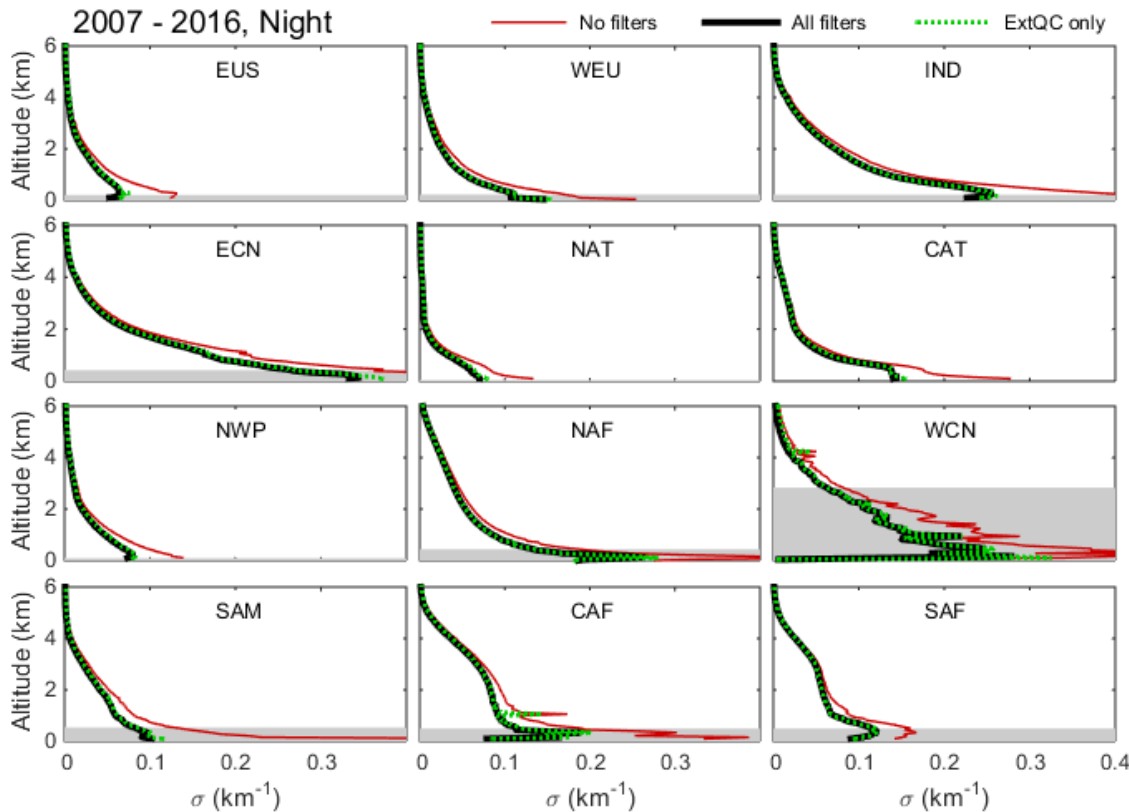

**Figure 22. Regional $\bar{\sigma}$ with no filters (red), all filters (black), and the extinction QC filter only (green dashed) for 2007–2016 at night, all-sky. The median surface elevation indicated by the shaded grey region.**

Regions experiencing the highest fractional AOD reductions also tend to have lower AOD both before and after filtering relative to other regions. For instance, the eastern United States (EUS) and north Atlantic Ocean (NAT) have the lowest annual AOD relative to other regions, yet the AOD reductions are among the highest: $\Delta$AOD = −34 % and −33 %, respectively (Fig. 23(a)). South America (SAM) experiences the highest annual AOD reduction, but this occurs during the March – May (MAM) season when mean AOD is lower than in the September – November (SON) season when biomass

burning becomes prevalent (AOD = 0.09 vs. 0.30, respectively; Table S1). A possible explanation is that the default $S_p$ for smoke is closer to the true value during the biomass burning season, requiring fewer $S_p$ reductions, compared to default $S_p$ values used during the non-burning DJF–MAM seasons. Mean AOD is reduced by around 24 % for regions with high AOD: India (IND), eastern China (ECN), north and central Africa (NAF, CAF). Mean AOD over western China (WCN) – the region with the highest AOD – is reduced by 34 %.

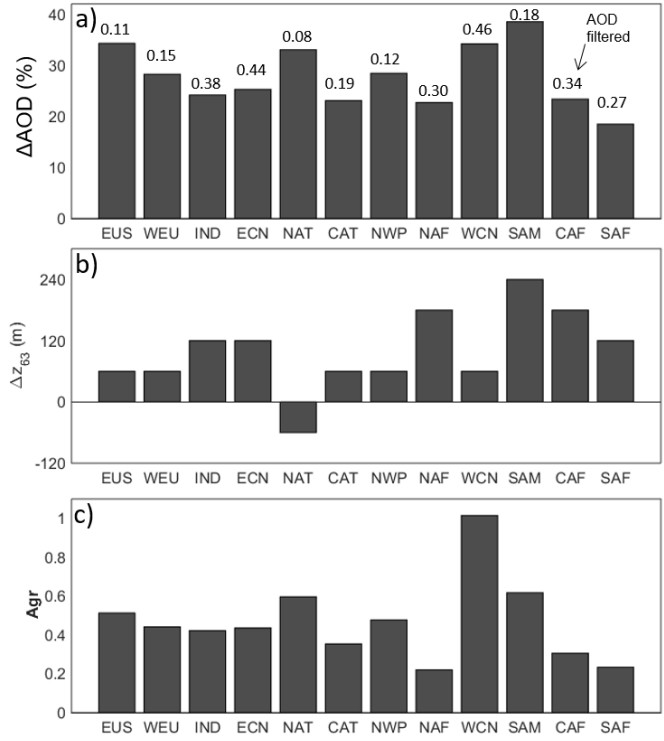

**Figure 23. Regional changes in AOD and $\bar{\sigma}$ with no filters compared with all filters: (a) percent reduction in AOD with numbers above the bars indicating mean filtered AOD, (b) difference in 63 % extinction scale heights (all filters – no filters; (B3)), and (c) filter aggressiveness (Eq. (B4)) for 2007–2016 at night, all-sky. Samples at altitudes ≤ 0.039 km are excluded due to low sample counts.**

The change in extinction scale height $\Delta z_{63}$ is positive for most regions (Fig. 23(b)), indicating that the altitude containing the bulk of the mean AOD is higher after quality filtering. For most regions, $z_{63}$ exhibits small increases of 60 m. Larger increases of 180–240 m occur over land where large $\sigma$ values near the surface are rejected. For example, the largest change occurs in SAM, where the maximum unfiltered extinction value at the surface was reduced substantially by quality filtering (Fig. 22). Since the majority of the AOD is no longer contained within the large near-surface peak, $z_{63}$ is higher after quality filtering. The CAF and southern Africa (SAF) regions also experience an $z_{63}$ increase of 120–180 m for this same reason.

Aerosol sample-weighted quality filter aggressiveness *Agr* (Fig. 23(c)) is largest in WCN, particularly during DJF (Fig. S7(c)), where AOD and the frequency of extQC = 2 solutions are relatively high (Fig. 8(e)). Since the extinction QC filter impacts a high proportion of the aerosol retrievals in this region, the filter has a strong impact on the mean aerosol extinction profile. Quality filtering is also relatively more aggressive in the NAT and northwest Pacific (NWP) regions where aerosol loading is typically low. When aerosol loading is low, rejecting just a small number of aerosol samples may have a large impact on $\bar{\sigma}$ because there are not many aerosol samples to begin with. Despite the substantial change to the SAF $\bar{\sigma}$ profile below the median surface elevation (Fig. 22), *Agr* is small relative to other regions because the metric is weighted by aerosol sample number, and most aerosol in that region is elevated above 1 km. Quality filtering is least aggressive over NAF.

## 7 Conclusions

The CALIOP level 3 aerosol profile product provides estimates of monthly mean globally gridded aerosol extinction profiles and AOD below 12 km, derived from CALIOP level 2 aerosol data. Given the uniqueness and length of this data record (over 10 years), it has been employed by the scientific community in numerous publications investigating the vertical distribution of aerosol. The quality filtering methods used in the level 3 product to minimize the influence of level 2 retrieval artifacts have also attracted interest from CALIOP data users. This paper thereby documents the quality filtering and averaging methods used to generate the level 3 aerosol profile product and serves as guidance for the use of CALIOP aerosol products.

In order to preserve the retrieved aerosol extinction profile shape, the level 3 algorithm aggregates extinction from the level 2 aerosol profile product rather than the level 2 aerosol layer product. Level 3 statistics are reported separately based on sky condition (i.e., cloud cover) to ensure versatility of possible applications: the cloud-free sky condition can be compared against measurements by passive sensors, all-sky maximizes sampling, and cloudy-sky statistics report solely what is missed by cloud-free observations. Day and night observations are reported separately to maintain similar levels of uncertainty and layer detection fidelity. In regions where no aerosol is detected by CALIOP (i.e., "clear-air" regions), aerosol extinction is assumed to be 0 km$^{-1}$. Thus, the mean aerosol extinction reported in the level 3 product represents a lower bound of the true aerosol extinction. The relative difference between the level 3 mean and the true aerosol extinction is expected to be least at altitudes where optically thick aerosol is most abundant (closer to the surface rather than the upper troposphere) and at night when level 2 layer detection is most successful at detecting optically thin features. Mean AOD is computed by first averaging quality-screened level 2 aerosol extinction profiles and then vertically integrating the result: i.e., average-then-integrate. This prevents a low bias that would result from alternately integrating the extinction profiles first and then averaging the set of AOD values (i.e., integrate-then-average).

Quality filters are applied to level 2 aerosol extinction profiles prior to aggregation in order to reduce the influence of layer detection errors, layer classification errors, extinction retrieval errors, and biases caused by the negative signal

anomaly. At low altitudes, the extinction QC flag filter is the most aggressive at changing the mean extinction profile and AOD. This filter prevents high-biases in mean aerosol extinction due to lidar ratio overestimates in regions where mixtures of multiple aerosol types require adjustments to the default lidar ratio (e.g., Arabian Sea and Saharan dust belt) or due to errors in retrieving overlying optical depth. Conversely, rejecting these layers causes an underestimate in occurrence frequency in these regions since these are likely legitimate aerosol layers. Suspected cloud contamination is reduced by the CAD score and misclassified cirrus fringe filters. At high altitudes, the cirrus fringe filter is the most aggressive at changing mean extinction, though the change in AOD is small.

Looking ahead, a new version of the level 3 aerosol product is under development which will ingest the version 4 level 2 data that was first released in November 2016. Version 4 level 2 benefits from a number of major improvements relevant to the level 3 aerosol product. Most significantly, updated aerosol lidar ratios and aerosol subtyping corrections (Kim et al., 2018) will have the largest impact on $\bar{\bar{\sigma}}$ and AOD reported by level 3 since these quantities are non-linear functions of lidar ratio (nearly linear at low optical depths). The overall structure of the level 3 aerosol profile product will remain similar in terms of grid size and science data sets, but the quality filtering strategy described in this paper may change to account for modifications in version 4 level 2 processing. Changes to future versions of the level 3 aerosol profile product will be documented by data quality summaries on the CALIPSO Data User's Guide website: https://www-calipso.larc.nasa.gov/resources/calipso_users_guide/

**Data Availability**

The CALIPSO lidar level 3 aerosol profile product is available for download from the Atmospheric Science Data Center at NASA Langley Research Center: https://eosweb.larc.nasa.gov/project/calipso/calipso_table.

**Appendix A Region definitions**

The latitude and longitude boundaries of regions discussed in Sect. 6.2 are defined in Table A1.

**Table A1. Regional latitude and longitude boundaries.**

| Region | | Lat. min | Lat. max | Long. min | Long. max |
|---|---|---|---|---|---|
| EUS | Eastern United States | 30° N | 48° N | 100° W | 70° W |
| WEU | Western Europe | 36° N | 58° N | 10° W | 50° E |
| IND | India | 6° N | 28° N | 70° E | 90° E |
| ECN | Eastern China | 20° N | 44° N | 105° E | 125° E |
| NAT | North Atlantic Ocean | 38° N | 54° N | 70° W | 30° W |
| CAT | Central Atlantic Ocean | 6° N | 34° N | 55° W | 20° W |
| NWP | Northwest Pacific Ocean | 32° N | 54° N | 125° E | 160° E |

| NAF | North Africa | 16° N | 34° N | 15° W | 60° E |
| WCN | Western China | 32° N | 44° N | 70° E | 100° E |
| SAM | South America | 24° S | 2° S | 75° W | 40° W |
| CAF | Central Africa | 0° N | 15° N | 15° W | 40° E |
| SAF | Southern Africa | 24° S | 2° S | 0° E | 45° E |

**Appendix B Mathematics of quality filtering metrics**

The following metrics are used in Sect. 6 to quantify the change in level 3 mean AOD and mean aerosol extinction due to quality filtering.

**B.1 Filter aggressiveness**

Filter aggressiveness is defined as the fractional change in mean extinction per fractional change in number of aerosol samples accepted due to quality filtering:

$$Agr(z) = \left| \frac{1 - \overline{\sigma}(z)_{filtered} / \overline{\sigma}(z)_{noFilters}}{1 - N(z)_{rejected} / N(z)_{aer,noFilters}} \right| \tag{B1}$$

Here, mean extinction values $\overline{\sigma}_{filtered}$ and $\overline{\sigma}_{noFilters}$ are computed with and without quality filters applied, and

sample statistics $N_{rejected}$ and $N_{aer,noFilters}$ indicate the number of aerosols rejected by quality filtering and the total number of aerosol samples prior to quality filtering, respectively. Large values of $Agr(z)$ indicate that quality filtering has changed mean level 3 extinction either by rejecting a large number of aerosol samples or by rejecting aerosol samples having large extinction.

**B.2 Extinction scale height**

The extinction scale height $z_{63}$ is defined as the altitude below which 63 % of the total mean AOD resides (Hayasaka et al., 2007):

$$\int_0^{z_{63}} \overline{\sigma}(z)\,dz = 0.63\,AOD = 0.63 \int_0^{12\,km} \overline{\sigma}(z)\,dz \tag{B2}$$

Here, $\overline{\sigma}$ is mean aerosol extinction and AOD is the total AOD integrated over the entire 12 km vertical extent. The extinction scale height difference used in Sect. 6 is defined as

$$\Delta z_{63} = z_{63,\,all\,filters} - z_{63,\,no\,filters} \tag{B3}$$

**B.3 Mean filter aggressiveness**

Filter aggressiveness $Agr(z)$ in Eq. (B1) is summarized for the entire mean aerosol extinction profile by computing the unfiltered aerosol sample-weighted mean of the impact metric. This incarnation gives **Agr** more weight to altitudes containing the most aerosol.

$$Agr = \frac{\sum N(z)_{aer,noFilters}\, Agr(z)}{\sum N(z)_{aer,noFilters}} \tag{B4}$$

**Supplement link**

**Competing interest**

The authors declare that they have no competing interests.

**Acknowledgements**

The authors thank the researchers who provided feedback to our team at the level 3 aerosol product peer review in September 2015. We would like to acknowledge Eleni Marinou and Vassilis Amiridis for recommending an improved strategy for single aerosol species averaging. We thank the CALIPSO Lidar Science Working Group for important feedback during product design, the CALIPSO Data Management Team for oversight of code development and curating the code base, and the Atmospheric Science Data Center at NASA Langley Research Center for archiving and hosting CALIPSO data. We appreciate the efforts of Charles Trepte and Patricia Lucker who managed the workforce that developed the level 3 aerosol product. We are grateful for the support of the CALIPSO mission provided by NASA Langley Research Center, Centre National d'Etudes Spatiales, and Science Systems and Applications, Inc. We would also like to thank the three anonymous reviewers who helped improve the manuscript.

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
