# Peer review of "CALIPSO lidar level 3 aerosol profile product: version 3 algorithm design"

_Atmospheric Measurement Techniques, 2018_

## Referee Comment (RC1) · Anonymous Referee #1 · 23 Apr 2018

This paper outlines the methodology behind the CALIPSO Level 3 data product. A simple average is used to aggregate Level 2 profiles, with clear skies assigned zero extinction (rather than being omitted from the average). The quality control system is described at some length, identifying unreliable data by geometrical, statistical, and algorithmic means (e.g. aerosols found in unexpected locations, a maximal uncertainty on extinction, and QC flags, respectively). The spatial and temporal impacts of each filter are presented, demonstrating that these choices alter the final result but in a manner that is expected to be more representative of reality.

The paper is suitable for publishing with only typographical corrections. The many years that went into its development are evident from the depth and extent of the discussions. Every detail is rationalised (and I eagerly await their justification in the

upcoming validation paper). Some familiarity with lidar is required to completely understand some of their choices (e.g. the fact that negative extinctions should be retained is less well known that it should be but neither cited nor discussed here), but that seems fair given the paper's length and audience. The language and presentation are exemplary throughout.

I include only a few very minor comments and corrections. P1L2 means line 2 of page 1.

§4.3 Considering aerosol properties tend to be log-normally distributed, have you explored averaging $\ln \sigma$?

P11L8 This recommendation seems sufficiently novel and important to be worth mentioning in the paper's abstract.

Figs. 6,7 Larger tick lengths in these colourbars could be beneficial.

P15L5 Add 'the' after 'During'.

P18L2 The peak of Fig. 13 looks nearer to 7 km than 6 to my eye.

P19L3 Delete the first 'the' and 'is'.

P19L3 What motivated the choice of 4 km as a floor for this filter rather than, say, 5 km?

P21L15 There should be commas before and after 'on the other hand'.

P21L30 Add 'the' before 'Antartic'.

Fig. 15 Any idea why (b) is stripy when all the other fields shown have been fairly smooth?

P23L4 I'd word this, '. . . higher altitudes because there are more likely to be overlying layers.'

P24L12 Delete the second 'or'.

Fig. 19 If this image only showed the region in the red box, it'd be quite convincing. As it stands, my attention is drawn to the difference between the spatial patterns of the two panes outside of your box, especially near the equator, and how the filter slightly reduces AOD in regions where the NSA is uncommon.

Fig. 22 Add 'the' before 'extinction'.

§A.1 The aggressivness metric is new to me. Have you adapted it from somewhere or invented it? Also, does it ever happen that the filter both rejects many samples and sharply decreases the extinction, which could result in small $Agr$?

P36L16 The page number for Koffi et al. 2012 is D10201.

P36L23 The page number for Koren et al. 2007 is L08805.

P38L20 The page number for Vernier et al. 2011 is L07804.

- On three on my devices with different operating systems, equations at P15L19 and P33L1 are poorly rendered with overlapping symbols.

- I believe the following should be hyphenated: P17L3 no-confidence; P18L11 and L13 high-confidence.

---

## Referee Comment (RC2) · Anonymous Referee #2 · 27 Apr 2018

The paper by Tackett *et al.* is very well written and provides the necessary information and guidance to construct the CALIPSO level 3 data product and, furthermore, to understand the ramifications of the quality screening procedure. The methodology employed for averaging extinction coefficient profiles and calculating the AOD as well as the filtering steps is clearly documented and accompanied by clear paradigms. The discussion on the impact of the quality screening criteria altogether and separately, both globally and regionally, demonstrates the choice of the relevant filtering steps. The paper reads very well and is appropriate for AMT. Minor revisions are given for consideration:

**Specific comments**

[Figure]

Pg3Ln28–34 & P4Ln1-6 & Fig.1: The choice of integrated aerosol information for constructing level 3 profiles sounds wrong to start with. The authors can keep the relevant discussion if they think it adds to the clarity of the document.

Pg6Ln19: Why not the other aerosol subtypes, in particular polluted continental?

Pg11Ln10: Could the differences between the version 3 level 3 quality screening strategy and the one reported in Winker et al. (2013) be described?

Pg19Ln3: A couple of lines could be spent to explain better how you arrived at the value of 4 km. What about higher latitudes?

Pg26 Fig.18: To my opinion, the area $160°–180°$ W and $20°–30°$ N should be boxed in as the effect of the filter is also evident.

Pg27 Fig.20: Will the "misclassified cirrus fringe filter" have the same impact for different height thresholds, smaller or greater to 4 km?

**Technical corrections**

Pg1Ln22: Add "the" before "version".

Pg2Ln29: Replace "335" with "333".

Pg3Ln24: Replace "a level 2" with "an".

Pg10Ln1 & Ln5: The triple bar could be avoided.

Pg10Ln8: Remove "Aerosol optical depth" and the parenthesis.

Pg10Ln8: Replace "passive satellites" with "spaceborne passive sensors".

Pg11Ln13–15: You can insert the subsection number to these main issues. For example, it can be "noise misclassified as aerosol (Sect. 5.1), clouds misclassified as aerosol (Sect. 5.2)".

Pg18Ln18: Fix the citation as "(Vernier et al., 2011, 2015)".

Pg19Ln3: Remove "the" before "having" and "is" before "less".

Pg21Ln26 & Ln28: The triple bars could be avoided.

Pg21Ln31: Remove ", however".

Pg23Ln4: I think something is missing after "likely" or consider removing "because overlying layers are more likely".

Pg23 Fig.16: Add the units for $\Delta\sigma$ in the caption.

Pg23Ln19: Give the acronym for ITCZ.

Pg25 Fig.18a: Add the colorbar.

Pg26 Fig.18b: What LL3 stands for?

Pg29Ln4: Add "shown in" before "Fig.S7".

Pg31Ln9: Replace "on" with "for" after "guidance".

Pg31Ln15: Add "of" after "levels".

Pg31Ln18: Add "the" after "and".
* * *

---

## Referee Comment (RC3) · Anonymous Referee #3 · 1 May 2018

The authors describe the Level 3 CALIOP monthly mean AOD and extinction profile products, their quality screening and averaging methods. This paper is of excellent quality and worthy of publication. It is very well written and very well structured.

Major comments: Any reference for these statements and assumptions would help the reader: . "The underlying assumption is that all aerosol layers below 250 m are in reality attached to the surface" . ". . . rarity of tropospheric aerosol detection above 12 km" . ". . . altitudes where shallow convective clouds are expected: above 8 km at the equator and lower towards the poles" . ". . . lidar ratio of a dust/marine mixture, which would fall in the range 20sr-40sr"

Please consider adding a Table in section 5 that summarizes all the quality filters that were applied to the level 2 data (i.e., iso80km, CAD, cirrusFringe, extQC, extUnc, NSA

filter).

It is not clear why the NSA filter is not part of Fig. 8 and 9

Fig. 18 b is not explained in the text. What is its purpose? Similar comment for Fig. 20b.

Consider giving two different names for variable "Agr" in Eq. A1 and Eq. A3.

Detailed comments:

P4 L23: "The 12 km upper limit was selected due to the rarity of tropospheric aerosol detection above 12 km in the level 2 product". Would it be possible to quantify that statement? CALIOP Level 3 would then be missing any aerosol event above 12km?

P5 L2: consider adding "at night" to "Figure 2 depicts these sky conditions for an individual level 2 granule" if that is the case (and consider adding that info in the legend of Fig. 2)

P6 L3: MODIS needs to be defined

P6 L13: consider adding "clouds" to "desert and snow"

P7 L16: this is if you consider well-mixed aerosols in an atmospheric column of 1km. Is that, right? If so, you might want to add this information.

P8 L10: "The underlying assumption is that all aerosol layers below 250 m are in reality attached to the surface". Is this assumption based on any observation? Any reference here, even case studies, would be helpful.

P8 L14: "Consequently, global mean level 3 AOD is increased by a small amount, roughly 1 %." This applies to this one case over the Arabian sea. Do you have more global statistics?

P11 L2: "sky" is repeated twice.

P11 L28-29: "Corresponding seasonal totals and averages, defined as December–

February (DJF), March–May (MAM), June–August (JJA) and September–November (SON), are also reported in supplementary material." Which figure(s) does this refer to?

Fig. 6 and 7: Are these gridded maps? 2° x 5°? Consider adding this information in the legend.

Fig. 8: Do "All filters" include the "NSA" filter as well (not shown in Fig. 8 bur in Fig. 19 instead)?

Fig. 8: consider adding the total percentage of aerosol samples rejected by each filter in the title of each of the 6 graphs

P15 L4: "altitudes where shallow convective clouds are expected: above 8 km at the equator and lower towards the poles". Any reference here?

P15 L5: delete "at"

P15 L6: Fig. S4b instead of Fig. S4a

P15 L30: "because there is no confidence in"

P16 L1-3: "Low CAD scores also indicate a high probability of layer detection artifacts where noise spikes cause the feature finder to detect layers that do not actually exist." However, the authors select data with CAD scores between -100 and -20. These are "low" CAD scores. Consider rephrasing.

Fig. 10: Is this figure global? If not, you might want to provide the latitude/ longitude range

P17 L22: "causing aerosol subtyping misclassifications". Is this the case for the aerosols near the surface in Fig.12? If so, are those corrected? It's not clear from the text.

P19 L3: "and having a cloud top temperature less than 0°C"

P19 L11: consider adding "in this case" to "The reduction in full column dust AOD..."

P19 L13: consider adding the dust AOD reduction in % in the text (i.e., values on Fig. 14)

P21 L3: "These are the least frequent of all solutions for aerosol layers, however" Can this be quantified?

P21 L28: "... lidar ratio of a dust/marine mixture, which would fall in the range ..." Consider adding references for typical lidar ratios.

P23 L17: consider rephrasing this way: "... profile shape the least while still rejecting solutions that are untrustworthy.

P23 L24: "while having only a small impact on mean AOD" Consider quantifying this statement

P24 L12: "or" is repeated twice

P24 L14: What do the authors mean by "this" example? Fig 18a?

Fig 18a is missing its color scale

Fig. 18b is not explained in the text (and Profile #1-3 are not described). What is its purpose?

Fig 19b: Consider adding AOD in the title.

P25 L6: "AOD increases by 5–10 % in regions affected by the NSA". The red box in 19b seems to show values between ∼0.8 and ∼1.2? The authors might refer to the pixels that show only a certain percentage of NSA frequency on 19a. if so, this needs to be specified. "AOD decreases by 5 % in unaffected regions" but it looks like it varies on 19b. More explanation here would be appreciated.

P26 L16: instead of "more aggressive at changing", consider "more impact than others"

Fig 20b: The description of this graph is not clear. It does not seem to be used in the

text either.

P27 L5: "dominate the aerosol sample rejection"

P27 L11: four instead of three metrics

P27 L12: delta_z_63 needs to be defined in the appendix, right after z_63 (eq. A2).

P27 L13: consider two different names for "Agr" in Eq. A1 and Eq. A3.

P27 L22: "For the remaining filters, delta_z_63 is either zero or decreases by 60 m because these filters act upon layers at higher altitudes". This is over ocean according to Table 2. You might want to describe what happens over land as well. Legend of Table 2: "and with no filters against all filters and with each filter applied independently". Consider rephrasing. Also, what is the reasoning behind the ocean and land separation in the Table?

Fig. 22: It is not clear why the authors show the surface elevation in this figure (not discussed in the text).

Fig. 23: Consider adding a mean AOD histogram as a fourth plot (instead of values on Fig 23a)

P30 L6: "greater geometric depth". Am I not understanding this correctly? Doesn't this mean z_63_filtered > z_63_non_filtered? Which would mean a higher altitude under which 63 % of the aerosols reside (instead of a geometric depth)?

P30 L16-18: "When aerosol loading is low, rejecting just a small number of aerosol samples may have a larger impact on sigma than in regions where aerosol loading is high since there are not many aerosol samples to begin with" This is not clear. Consider rephrasing.

P31 L16: similar levels of uncertainty

P31 L24: in order to reduce

[Figure]

Appendix: Table B1 comes sooner in the text than appendix A: consider switching Appendix A and B.

Fig. S1-S2 consider "quality screening" instead of "data filtering" to match the legend of Fig. 6-7

[Figure]

---

## Author Comment (AC1) · 27 Jun 2018

**Response to Anonymous Referee #1**

We thank the referee for their helpful comments, suggestions, and for pointing out typos. They have helped improve the manuscript substantially. Referee comments and original text are shown below in black font. Responses and modifications to text are show in red font.

This paper outlines the methodology behind the CALIPSO Level 3 data product. A simple average is used to aggregate Level 2 profiles, with clear skies assigned zero extinction (rather than being omitted from the average). The quality control system is described at some length, identifying unreliable data by geometrical, statistical, and algorithmic means (e.g. aerosols found in unexpected locations, a maximal uncertainty on extinction, and QC flags, respectively). The spatial and temporal impacts of each filter are presented, demonstrating that these choices alter the final result but in a manner that is expected to be more representative of reality.

The paper is suitable for publishing with only typographical corrections. The many years that went into its development are evident from the depth and extent of the discussions. Every detail is rationalised (and I eagerly await their justification in the
upcoming validation paper). Some familiarity with lidar is required to completely understand some of their choices (e.g. the fact that negative extinctions should be retained is less well known that it should be but neither cited nor discussed here), but that seems fair given the paper's length and audience. The language and presentation are exemplary throughout.

I include only a few very minor comments and corrections. P1L2 means line 2 of page 1.
* * *
§4.3 Considering aerosol properties tend to be log-normally distributed, have you explored averaging ln σ?

We felt a simple average would be appeal to a broad range of applications so we never explored averaging ln σ. We also believe it is important to retain the negative extinctions reported by CALIOP and "clear-air" extinction values of 0.0 /km in the average, which cannot be accomplished by taking the logarithm of extinction.
* * *
P11L8 This recommendation seems sufficiently novel and important to be worth mentioning in the paper's abstract.

Great suggestion. The abstract has been revised as follows, with new text in red.

> *The impact of quality screening on monthly mean aerosol extinction is investigated globally and regionally. After applying quality filters, the level 3 algorithm calculates monthly mean AOD by vertically integrating the monthly mean quality-screened aerosol extinction profile. Calculating monthly mean AOD by integrating the monthly mean extinction profile prevents a low bias that would result from alternately integrating the set of extinction profiles first and then averaging the resultant AOD values together. Ultimately, the quality filters reduce level 3 mean AOD by −24 and −31 % for global ocean and global land, respectively, indicating the importance of quality screening.*
* * *
Figs. 6,7 Larger tick lengths in these colourbars could be beneficial.

Done
* * *
P15L5 Add 'the' after 'During'.

Done
* * *
P18L2 The peak of Fig. 13 looks nearer to 7 km than 6 to my eye.

Agreed. The text has been changed from 6 km to 7 km.
* * *
P19L3 Delete the first 'the' and 'is'.

Done
* * *
P19L3 What motivated the choice of 4 km as a floor for this filter rather than, say, 5 km?

The following text was added to the manuscript to provide a rationale for the 4 km threshold. Within $4 \pm 1$ km, the choice is somewhat arbitrary. The intention was to choose an altitude which encompassed the bulk of aerosol retrievals and where we know that ice clouds also exist.

> *The 4 km altitude threshold limits the magnitude of error that would be made by rejecting legitimate aerosol in the lower troposphere where aerosol and clouds are more likely to coexist. For example, 95 % of all aerosol layers detected in 2010 are below 4 km (global). Meanwhile, 11 % of all ice clouds are also detected below this altitude. Ice clouds below 4 km are even more frequent at high latitudes: comprising ~22 % of all ice clouds at latitudes higher than 50° N/S in 2010. The global 4 km threshold thereby protects the majority of legitimate aerosols from being incorrectly rejected, albeit with the possibility of some remaining cirrus fringes at high latitudes.*
* * *
P21L15 There should be commas before and after 'on the other hand'.

Done
* * *
P21L30 Add 'the' before 'Antartic'.

Done

Fig. 15 Any idea why (b) is stripy when all the other fields shown have been fairly smooth?

The stripes in this figure occur at high altitudes, primarily over the Antarctic. This is because the number of aerosol samples is very low in these regions by several orders of magnitude relative to elsewhere (Fig. R1) which causes noisiness in the frequency calculation.

[Figure]

Figure R1. Number of aerosol samples for 2007-2016 at night, cloud-free with no filters applied.
* * *
P23L4 I'd word this, '. . . higher altitudes because there are more likely to be overlying layers.

Done. I like your recommendation.
* * *
P24L12 Delete the second 'or'.

Done
* * *
Fig. 19 If this image only showed the region in the red box, it'd be quite convincing. As it stands, my attention is drawn to the difference between the spatial patterns of the two panes outside of your box, especially near the equator, and how the filter slightly reduces AOD in regions where the NSA is uncommon.

These are good points. The difference in spatial patterns adds to the rationale for using the filter (it is hard to predict the influence of the NSA) while the reduction in AOD where the NSA is uncommon is worth emphasizing. We also want to note that the location of the NSA varies with season. The text has been revised in this section to reflect these issues (P24L9, P25L6 in original manuscript). The new text is shown in red below.

> *The NSA is intermittent, but tends to occur in sequences of adjacent profiles within latitude bands that vary seasonally.*
> *…*
> *An example of the impact of this NSA mitigation is shown in Fig. 19. AOD increases by roughly 5–10 % in level 3 profiles affected by the NSA (based on values > 1 within the red boxes) because strongly negative near-surface $\sigma$ is rejected. Conversely, the NSA also is present in this example along the*

*equator, and yet excluding these σ values does not increase AOD, illustrating the difficulty of predicting the influence of the NSA on retrieved extinction. AOD also decreases by roughly 5 % on average in unaffected regions, a consequence of this conservative strategy.*
* * *
Fig. 22 Add 'the' before 'extinction'.

Done
* * *
§A.1 The aggressivness metric is new to me. Have you adapted it from somewhere or invented it? Also, does it ever happen that the filter both rejects many samples and sharply decreases the extinction, which could result in small Agr?

The aggressiveness metric was invented for this paper. Yes, this is possible, but in that case the filter was not particularly aggressive because it took rejecting many samples to decrease the extinction (even though it decreased by a lot). A small value of Agr would be appropriate based on this interpretation of "aggressiveness".
* * *
P36L16 The page number for Koffi et al. 2012 is D10201.
P36L23 The page number for Koren et al. 2007 is L08805.
P38L20 The page number for Vernier et al. 2011 is L07804.

All three citations are now correct. Thank you.
* * *
• On three on my devices with different operating systems, equations at P15L19 and P33L1 are poorly rendered with overlapping symbols.

I will bring this comment to the attention of the AMT typesetting editor to ensure that these equations are rendered properly in the final publication. For your convenience, these lines are replicated below as screenshots.

> The cloud-aerosol discrimination (CAD) algorithm evaluates five CALIOP observables to classify layers as aerosol or cloud: 532 nm layer-mean attenuated backscatter ($< \beta'_{532} >$), layer-mean attenuated color ratio ($\chi' = < \beta'_{1064} > / < \beta'_{532} >$),
>
> 20  layer-integrated volume depolarization ratio ($\delta_v$), latitude, and altitude. These five observables are evaluated against five

$$\int_0^{z_{63}} \overline{\sigma}(z)\, dz = 0.63\, AOD = 0.63 \int_0^{12\,km} \overline{\sigma}(z)\, dz \tag{A2}$$

> Here, $\overline{\sigma}$ is mean aerosol extinction and AOD is the total AOD integrated over the entire 12 km vertical extent.
* * *
• I believe the following should be hyphenated: P17L3 no-confidence; P18L11 and L13 high-confidence.

Agreed. The hyphens have been added.

---

## Author Comment (AC2) · 27 Jun 2018

Response to Anonymous Referee #2

We thank the referee for the useful comments, suggestions, and corrections. Addressing the questions posed by the referee have helped us to improve the clarity of the manuscript. Referee comments and original text are shown in black font below. Responses to referee comments and modified text are shown in red font. Page and line numbers below refer to the original manuscript.

The paper by Tackett et al. is very well written and provides the necessary information and guidance to construct the CALIPSO level 3 data product and, furthermore, to understand the ramifications of the quality screening procedure. The methodology employed for averaging extinction coefficient profiles and calculating the AOD as well as the filtering steps is clearly documented and accompanied by clear paradigms. The discussion on the impact of the quality screening criteria altogether and separately, both globally and regionally, demonstrates the choice of the relevant filtering steps. The paper reads very well and is appropriate for AMT. Minor revisions are given for consideration:

Specific comments

Pg3Ln28–34 & P4Ln1-6 & Fig.1: The choice of integrated aerosol information for constructing level 3 profiles sounds wrong to start with. The authors can keep the relevant discussion if they think it adds to the clarity of the document.

Our team has worked with several data users to help them understand the importance of using the level 2 profile product rather than level 2 layer product to generate profile information. We sense that this is a point of confusion with using CALIOP products to generate level 3-style averages. We believe this paper is an appropriate opportunity to clarify how to properly use the profile data and to demonstrate the ill effects that would arise by using the layer data. We attempted to keep the discussion on this topic brief.

<hr>

Pg6Ln19: Why not the other aerosol subtypes, in particular polluted continental?

In order to manage the file size of the level 3 product, we chose to include a subset of aerosol subtypes. CALIOP excels at dust detection so dust and polluted dust were prime candidates. Since polluted dust is intended to be a mixture of smoke and dust, it made sense to include it (and CALIOP does well at detecting elevated smoke). We chose to stick with those three subtypes to keep the file size from becoming too large. If we receive requests from the science community desiring polluted continental, we may consider including it in a future version of the level 3 product.

<hr>

Pg11Ln10: Could the differences between the version 3 level 3 quality screening strategy and the one reported in Winker et al. (2013) be described?

Yes, the following text in red was added to the introduction of Section 5.

These filters are designed to counteract four main issues: noise misclassified as aerosol, clouds misclassified as aerosol, extinction retrieval errors, and an instrument artifact that intermittently produces large negative signals near the surface. All of these filters, except the last, are identical to

1

*filters A1 – A5 described in Appendix A of Winker et al. (2013) for the beta level 3 product. The near-surface negative signal anomaly filter (Sect. 5.4) replaces filter A6 of Winker et al. (2013).*

The following text in red was also added to the end of Sect. 4.2 (Pg8Ln11 in the original manuscript):

*To avoid a low bias in near-surface mean aerosol extinction, the level 3 algorithm ignores all clear-air samples below the lowest aerosol layer in each column having a base below 250 m....[Note that the beta version of the level 3 product used 2.46 km rather than 250 m as the threshold (Winker et al., 2013)].*
* * *
Pg19Ln3: A couple of lines could be spent to explain better how you arrived at the value of 4 km. What about higher latitudes?

The following sentences in red were added to the manuscript to provide a rationale for the 4 km threshold.

*The 4 km altitude threshold limits the magnitude of error that would be made by rejecting legitimate aerosol in the lower troposphere where aerosol and clouds are more likely to coexist. For example, 95 % of all aerosol layers detected in 2010 are below 4 km (global). Meanwhile, 11 % of all ice clouds are also detected below this altitude. Ice clouds below 4 km are even more frequent at high latitudes: comprising ~22 % of all ice clouds at latitudes higher than 50° N/S in 2010. The global 4 km threshold thereby protects the majority of legitimate aerosols from being incorrectly rejected, albeit with the possibility of some remaining cirrus fringes at high latitudes.*
* * *
Pg26 Fig.18: To my opinion, the area 160∘–180∘ W and 20∘–30∘ N should be boxed in as the effect of the filter is also evident.

This comment refers to Fig. 19. The additional boxes have been added.
* * *
Pg27 Fig.20: Will the "misclassified cirrus fringe filter" have the same impact for different height thresholds, smaller or greater to 4 km?

The fringe filter would remain the most aggressive filter at the highest altitudes, say above 6 km (Fig 20(d)) if the altitude threshold was adjusted up or down by a kilometer. The frequency of sample rejection (Fig. 20(c)) would look the same, but with a cutoff altitude corresponding to the new altitude threshold. The impact on global mean AOD and the mean extinction profiles would be small because any additional sample rejected or accepted due to the threshold moving up or down by a kilometer or so would mostly have small extinction values (Fig 20(a)). I modified the 4 – 5 km value in following line to "~6 km" to better state where the cirrus fringe filter is the most aggressive (Pg27 Ln. 8):

*Above ~6 km, the cirrus fringe filter is by far the most aggressive at changing $\bar{\bar{\sigma}}$. A similar conclusion is expected for small (~ 1 km) perturbations of the 4 km altitude threshold for this filter.*

**Technical corrections**

Pg1Ln22: Add "the" before "version".

Done

Pg2Ln29: Replace "335" with "333".

Done

Pg3Ln24: Replace "a level 2" with "an".

Done
* * *
Pg10Ln1 & Ln5: The triple bar could be avoided

I prefer to keep the triple bar to emphasize that the extinction value of 0 /km is assigned rather than retrieved or measured. To explain the meaning of the notation, the following in red was added to Pg7Ln10:

> *Level 2 atmospheric samples classified as "clear-air" (i.e., no feature is detected) are assumed in the level 3 algorithm to have aerosol extinction equal to 0 $km^{-1}$, denoted by $\sigma_{clear}$ (specifically, extinction $\sigma$ for clear-air samples are assigned $\sigma \equiv \sigma_{clear}$ ; the triple bar denotes the assignment).*

I also replaced "setting" with "assigning" on Pg10Ln1 & Ln5 to be consistent with the language above (i.e., inferring to the reader that the triple bar denotes an assignment).

> *Therefore, assigning $\sigma_{aer} \equiv 0\ km^{-1}$ for other species is equivalent to assuming that only one aerosol type is present in the detected layer…Assigning $\sigma_{aer} \equiv 0\ km^{-1}$ for other species avoids these biases and maintains consistency with the CALIPSO aerosol typing paradigm.*
* * *
Pg10Ln8: Remove "Aerosol optical depth" and the parenthesis.

Done
* * *
Pg10Ln8: Replace "passive satellites" with "spaceborne passive sensors".

Done
* * *
Pg11Ln13–15: You can insert the subsection number to these main issues. For example, it can be "noise misclassified as aerosol (Sect. 5.1), clouds misclassified as aerosol (Sect. 5.2)".

Done. That is a superb idea.
* * *
Pg18Ln18: Fix the citation as "(Vernier et al., 2011, 2015)".

Done
* * *
Pg19Ln3: Remove "the" before "having" and "is" before "less".

Done
* * *
Pg21Ln26 & Ln28: The triple bars could be avoided.

I prefer to keep the triple bars to emphasize that these are the default lidar ratio values assigned by the lidar ratio selection algorithm. Granted, the assigned values are meant to represent the actual lidar ratios observed in nature, but there is some natural variability in lidar ratios. I want to be clear that the errors discussed in this paragraph are due to differences in the assigned values. To explain the notation, the following text in red was added:

> *In this region, dust ($S_p \equiv 40\ sr$) commonly mixes with marine aerosol ($S_p \equiv 20\ sr$) and this mixture is misclassified as polluted dust by the version 3 aerosol typing algorithm (the triple bars denote that these are default assigned values).*
* * *
Pg21Ln31: Remove ", however".

"However" has been moved to earlier in the sentence so it reads more smoothly:

> *Rejections over the Antarctic, however, are more often caused by special error states listed in Table 1 rather than the need to adjust $S_p$.*
* * *
Pg23Ln4: I think something is missing after "likely" or consider removing "because overlying layers are more likely".

The sentence has been rephrased to:

> *Aerosol layers near the surface therefore tend to have larger $\Delta\sigma$ compared to those at higher altitudes because there are more likely to be overlying layers.*
* * *
Pg23 Fig.16: Add the units for $\Delta\sigma$ in the caption.

Done
* * *
Pg23Ln19: Give the acronym for ITCZ.

Replaced "ITCZ: with "intertropical convergence zone".
* * *
Pg25 Fig.18a: Add the colorbar.

Done
* * *
Pg26 Fig.18b: What LL3 stands for?

This comment refers to Fig. 19b which has "LL3" in the title. "LL3" stands for "lidar level 3", referring to the product. However, to improve clarity, I removed "LL3" and replaced the title of this panel to "AOD ratio (filter/no filter)".
* * *
Pg29Ln4: Add "shown in" before "Fig.S7".

Done
* * *
Pg31Ln9: Replace "on" with "for" after "guidance".

Done. A similar statement on Pg2Ln16 was also changed to use "for".
* * *
Pg31Ln15: Add "of" after "levels".

Done
* * *
Pg31Ln18: Add "the" after "and".

Done

---

## Author Comment (AC3) · 27 Jun 2018

**Response to Anonymous Referee #3**

We thank the referee for the many useful comments, questions, suggestions, and typographical corrections. They have helped improve the clarity of the manuscript. Referee comments and original text are shown in black font below. Reponses to referee comments and revised text are shown in red font. Page numbers and line numbers refer to the original manuscript.

The authors describe the Level 3 CALIOP monthly mean AOD and extinction profile products, their quality screening and averaging methods. This paper is of excellent quality and worthy of publication. It is very well written and very well structured. Major comments: Any reference for these statements and assumptions would help the reader: . "The underlying assumption is that all aerosol layers below 250 m are in reality attached to the surface" . ". . . rarity of tropospheric aerosol detection above 12 km" . ". . . altitudes where shallow convective clouds are expected: above 8 km at the equator and lower towards the poles" . ". . . lidar ratio of a dust/marine mixture, which would fall in the range 20sr-40sr" Please consider adding a Table in section 5 that summarizes all the quality filters that were applied to the level 2 data (i.e., iso80km, CAD, cirrusFringe, extQC, extUnc, NSA filter).

The requested references have been added and are discussed in the detailed responses below. We agree that adding a table to summarize the quality filters is useful for readers, given that several other publications which apply quality filtering to CALIOP data do the same. The table has been added to the end of Sect. 3 (replicated below). We chose this location because it is necessary to list items discussed in Sect. 4 to fully describe methods used to generate level 3 output.

*Table 1 is given here as a high-level summary of the averaging methods and quality filtering procedures detailed in the following two sections.*

**Table 1. Summary of averaging methods and quality filtering procedures used to generate the version 3 level 3 aerosol product. Details are discussed in the indicated sections. AGL and AMSL indicate "above ground level" and "above mean sea level", respectively.**

| Averaging method / quality filtering procedure | Section |
|---|---|
| Aerosol extinction for "clear-air" assigned $\equiv$ 0 km$^{-1}$ | 4.1 |
| Clear-air below aerosol layers with bases < 250 m AGL ignored | 4.2 |
| Isolated 80 km horizontal resolution aerosol layers rejected | 5.1 |
| CAD score outside [−100, −20] range rejected | 5.2.1 |
| Aerosol in contact with ice clouds (top temperature < 0° C) above 4 km AMSL rejected | 5.2.2 |
| Extinction QC flag $\neq$ 0, 1, 16, 18 rejected | 5.3.1 |
| Extinction uncertainty = 99.9 km$^{-1}$ rejected, and all extinction below | 5.3.2 |
| All samples $\leq$ 60 m AGL excluded | 5.4 |

It is not clear why the NSA filter is not part of Fig. 8 and 9

The NSA filter ignores all information within 60 m of the surface so any aerosol samples are not eligible for rejection. Hence, there is no frequency of rejection to report in Fig. 8 or 9.

Fig. 18 b is not explained in the text. What is its purpose? Similar comment for Fig. 20b.

Figure 18b is explained on P24 L13:

> *Figure 18(b) shows three aerosol extinction profiles retrieved from the attenuated backscatter in Fig 18(a). While the strongly negative values adjacent to the surface are readily apparent in this example, positive $\sigma$ values that are biased low are not as easy to detect.*

Figure 20b shows the number of unfiltered aerosol samples. It is shown to provide context for the number of samples contributing to the other three panels in the figure. The following sentence in red was added to the manuscript on P26 L17.

> *Figure 20 summarizes the impact of quality filters on the global $\bar{\sigma}$ profile, the frequency of rejection, and filter aggressiveness. ... For context, Fig. 20(b) shows the number of unfiltered aerosol samples, which decreases with increasing altitude.*
* * *
Consider giving two different names for variable "Agr" in Eq. A1 and Eq. A3.

Thank you for the suggestion. We find the two variations, $Agr(z)$ and **Agr** (now Eq. B1 and Eq. B3), work well in terms of distinguishability and consistency so we will retain the current names.
* * *
Detailed comments:

P4 L23: "The 12 km upper limit was selected due to the rarity of tropospheric aerosol detection above 12 km in the level 2 product". Would it be possible to quantify that statement? CALIOP Level 3 would then be missing any aerosol event above 12km?

The following statement in red was added to quantify the statement. Yes, CALIOP level 3 will miss these high-altitude events, but this is acceptable because the emphasis for this product is the lower troposphere.

> *The 12 km upper limit was selected due to the rarity of tropospheric aerosol detection above 12 km in the level 2 product (e.g., 0.04 % of tropospheric aerosol layers detected by CALIOP version 3 are above 12 km in 2010).*
* * *
P5 L2: consider adding "at night" to "Figure 2 depicts these sky conditions for an individual level 2 granule" if that is the case (and consider adding that info in the legend of Fig. 2)

The "nighttime granule" qualifier is now added to the Fig. 2 caption, but not to the main text because the day/night detail is not pertinent to the describing how the figure conveys the sky conditions.
* * *
P6 L3: MODIS needs to be defined

Done
* * *
P6 L13: consider adding "clouds" to "desert and snow"

Added the text in red:

> *In daytime, the signal-to-noise ratio (SNR) is lower relative to night, particularly over high albedo surfaces such as desert or snow or over clouds (Hunt et al., 2009).*
* * *
P7 L16: this is if you consider well-mixed aerosols in an atmospheric column of 1km. Is that, right? If so, you might want to add this information.

It assumes well-mixed aerosols in an atmospheric column of 10 km. The following statement in red was added:

> *Clarke and Kapustin (2002), for example, show background aerosol extinction levels of $10^{-4}$ $km^{-1}$ to $10^{-3}$ $km^{-1}$ in remote parts of the Pacific basin, implying a missing AOD ranging from $10^{-3}$ to $10^{-2}$ in the cleanest regions (assuming well-mixed aerosols in a 10 km deep column).*
* * *
P8 L10: "The underlying assumption is that all aerosol layers below 250 m are in reality attached to the surface". Is this assumption based on any observation? Any reference here, even case studies, would be helpful.

The underlying assumption is that aerosol is well mixed within the planetary boundary layer at these altitudes. The sentence was rephrased to better express this assumption. Two references are given as examples which retrieved planetary boundary layer (PBL) heights from CALIOP data. Figure 3 of McGrath-Spangler and Denning (2013) shows maps of seasonal mean daytime PBL depths which are primarily above 500 m. Figures 3 and 7 of Luo et al., (2014) show PBL heights (labelled as BLH) for marine and continental sites also above 250 m.

> *The underlying assumption is that the atmosphere is well mixed below 250 m. Turbulent mixing within the daytime boundary layer tends to homogenize aerosol loading, and the planetary boundary layer is generally much deeper than 250 m for marine and continental conditions (e.g., McGrath-Spangler and Denning (2013); Luo et al. (2014)).*
* * *
P8 L14: "Consequently, global mean level 3 AOD is increased by a small amount, roughly 1 %." This applies to this one case over the Arabian sea. Do you have more global statistics?

The AOD increase of 1 % is the global mean value, not just for the Arabian Sea case.

P11 L2: "sky" is repeated twice.

Corrected
* * *
P11 L28-29: "Corresponding seasonal totals and averages, defined as December–February (DJF), March–May (MAM), June–August (JJA) and September–November (SON), are also reported in supplementary material." Which figure(s) does this refer to?

These figures are not included in the supplementary material. This line has been removed in the revised manuscript. Seasonal acronym definitions now appear as they are used.
* * *
Fig. 6 and 7: Are these gridded maps? 2∘ x 5∘? Consider adding this information in the legend.

These are maps of statistics reported by the level 3 product so they are 2 x 5 degree resolution. The legends for Figs 6-8 and Figs S1- S4 now include "as reported by the level 3 product" to make this clear.
* * *
Fig. 8: Do "All filters" include the "NSA" filter as well (not shown in Fig. 8 bur in Fig. 19 instead)?

Yes, the NSA filter is applied to all figures except in Fig. 19 which demonstrates the effect. The NSA filter is different than other filters because no samples are rejected, they are just ignored within 60 m of the surface at all times.
* * *
Fig. 8: consider adding the total percentage of aerosol samples rejected by each filter in the title of each of the 6 graphs

This is a very good idea. The percentages have been added to Figs. 8 and S3.
* * *
P15 L4: "altitudes where shallow convective clouds are expected: above 8 km at the equator and lower towards the poles". Any reference here?

Added the following reference which shows typical altitudes of deep convection based on combined CloudSat/CALIPSO data. Also, replaced "shallow convective clouds" with "deep convective clouds" because that is the cloud type being discussed.

> Mace, G.G. and F.J. Wrenn, 2013: Evaluation of the Hydrometeor Layers in the East and West Pacific within ISCCP Cloud-Top Pressure–Optical Depth Bins Using Merged CloudSat and CALIPSO Data. J. Climate, 26, 9429–9444, https://doi.org/10.1175/JCLI-D-12-00207.1
* * *
P15 L5: delete "at"

Done
* * *
P15 L6: Fig. S4b instead of Fig. S4a

Corrected
* * *
P15 L30: "because there is no confidence in"

Corrected
* * *
P16 L1-3: "Low CAD scores also indicate a high probability of layer detection artifacts where noise spikes cause the feature finder to detect layers that do not actually exist." However, the authors select data with CAD scores between -100 and -20. These are "low" CAD scores. Consider rephrasing.

Indeed, the sentence is discussing no-confidence CAD scores rather than low-confidence CAD scores. The following changes were made to clarify.

> *A CAD score of −100 indicates that the feature is very likely an aerosol layer, and a CAD score of +100 indicates that the feature is very likely a cloud. There is no confidence in cloud-aerosol discrimination for features with |CAD score| < 20. For the year 2010 at night in version 3, over 85 % of aerosol layers have CAD score < −90 and around 4 % have CAD score > −20. The remaining 11 % have intermediate levels of confidence.*
>
> *Aerosol layers having CAD scores outside the range of [−100, −20] are rejected because there is no confidence in discriminating aerosol from cloud... No-confidence CAD scores also indicate a high probability of layer detection artifacts where noise spikes cause the feature finder to detect layers that do not actually exist.*
* * *
Fig. 10: Is this figure global? If not, you might want to provide the latitude/ longitude range

Amended the figure caption:

> *Figure 10. Median overlying integrated attenuated backscatter (IAB) for aerosol layers having the indicated CAD score for 2010 at night, global.*
* * *
P17 L22: "causing aerosol subtyping misclassifications". Is this the case for the aerosols near the surface in Fig.12? If so, are those corrected? It's not clear from the text.

Yes, this explains why the marine aerosol is misclassified as dust near the surface in Fig. 12. These misclassifications are not corrected because the cirrus fringe filter is only applied above 4 km. The text has been revised as shown below to clarify.

> *The aerosol classified as dust within the marine boundary layer (albeit infrequently, < 0.3 %) is likely associated with residual cloud layers detected at 1/3 km resolution affecting $\delta_v$, causing aerosol subtyping misclassifications. However, the enhanced frequency of dust detection at higher altitudes is the main issue addressed by this filter: when the cirrus fringe filter is not applied (blue profile), the peak altitude of dust frequency appears at nearly 7 km.*
* * *
P19 L3: "and having a cloud top temperature less than 0∘C"

Corrected
* * *
P19 L11: consider adding "in this case" to "The reduction in full column dust AOD. . ."

Added
* * *
P19 L13: consider adding the dust AOD reduction in % in the text (i.e., values on Fig. 14)

The percent dust AOD reduction for Fig. 14 is given in the text on P19 L11:

> *"The reduction in full column dust AOD is small, about 7 %)."*
* * *
P21 L3: "These are the least frequent of all solutions for aerosol layers, however" Can this be quantified?

The following text in red was added with the percentage given in Table 1: *"These are the least frequent of all solutions for aerosol layers, however (~0.01 % of all retrievals)."*
* * *
P21 L28: ". . . lidar ratio of a dust/marine mixture, which would fall in the range . . ." Consider adding references for typical lidar ratios.

Added a citation to Omar et al. 2018 which discusses the CALIOP lidar ratio used for dust/marine mixtures and provides references to relevant CALIOP papers discussing typical lidar ratio values.
* * *
P23 L17: consider rephrasing this way: ". . . profile shape the least while still rejecting solutions that are untrustworthy.

Done. Good call.

P23 L24: "while having only a small impact on mean AOD" Consider quantifying this statement

The following in red was added: *"...while having only a small impact on global mean AOD (a reduction of ~10 %)."*
* * *
P24 L12: "or" is repeated twice

Corrected
* * *
P24 L14: What do the authors mean by "this" example? Fig 18a?

It refers to Fig. 18(b). The text was modified to clarify that the example is the extinction profiles (not the attenuated backscatter): *"While the strongly negative values adjacent to the surface are readily apparent for the extinction profiles in this example,…"*
* * *
Fig 18a is missing its color scale

The color scale is now added.
* * *
Fig. 18b is not explained in the text (and Profile #1-3 are not described). What is its purpose?

The description of this figure is revised as shown below. The extinction profiles show how the NSA in level 1B attenuated backscatter affects level 2 aerosol extinction retrievals by causing a strongly negative value adjacent to the surface. These are three extinction profiles

> *Figure 18(b) shows three aerosol extinction profiles retrieved from the attenuated backscatter in Fig 18(a) along separate 5 km segments containing the NSA. While the strongly negative values adjacent to the surface are readily apparent for the extinction profiles in this example, positive $\sigma$ values that are biased low are not as easy to detect.*
* * *
Fig 19b: Consider adding AOD in the title.

The title has been re-labeled as "AOD ratio (filter/no filter)"
* * *
P25 L6: "AOD increases by 5–10 % in regions affected by the NSA". The red box in 19b seems to show values between ~0.8 and ~1.2? The authors might refer to the pixels that show only a certain percentage of NSA

frequency on 19a. if so, this needs to be specified. "AOD decreases by 5 % in unaffected regions" but it looks like it varies on 19b. More explanation here would be appreciated.

The language below in red was added to clarify that the 5 – 10% values are rough estimates based only on the values > 1 within the red boxes. The 5 % value is based on the average.

> *AOD increases by roughly 5–10 % in level 3 profiles affected by the NSA (based on values > 1 within the red boxes) because strongly negative near-surface $\sigma$ is rejected. Conversely, the NSA is also present in this example along the equator, and yet excluding these $\sigma$ values does not increase AOD, illustrating the difficulty of predicting the influence of the NSA on retrieved extinction. AOD also decreases by roughly 5 % on average in unaffected regions, a consequence of this conservative strategy.*
* * *
P26 L16: instead of "more aggressive at changing", consider "more impact than others"

I prefer to use "aggressive" to maintain consistent language throughout this section when referring to this filter. The wording in red below was added to help clarify the statement, however.

> *The aggressiveness metric $Agr(z)$ indicates the effectiveness of sample rejection on changing $\bar{\sigma}$, with larger values indicating the filter is more aggressive at changing $\bar{\sigma}$ than other filters.*
* * *
Fig 20b: The description of this graph is not clear. It does not seem to be used in the text either.

As commented previously, Figure 20b shows the number of unfiltered aerosol samples. It is shown to provide context for the number of samples contributing to the other three panels in the figure. The following sentence in red was added to the manuscript on P26 L17.

> *Figure 20 summarizes the impact of quality filters on the global $\bar{\sigma}$ profile, the frequency of rejection, and filter aggressiveness. ... For context, Fig. 20(b) shows the number of unfiltered aerosol samples, which decreases with increasing altitude.*
* * *
P27 L5: "dominate the aerosol sample rejection"

Done
* * *
P27 L11: four instead of three metrics

Corrected
* * *
P27 L12: delta_z_63 needs to be defined in the appendix, right after z_63 (eq. A2).

The following was added to Appendix A (now Appendix B):

> *The extinction scale height difference used in Sect. 6 is defined as*
>
> $$\Delta z_{63} = z_{63,\,all\,filters} - z_{63,\,no\,filters} \qquad\qquad (B3)$$
* * *
P27 L13: consider two different names for "Agr" in Eq. A1 and Eq. A3.

Thank you for the suggestion. We find the two variations, $Agr(z)$ and **Agr**, work well in terms of distinguishability and consistency so we will retain the current names.
* * *
P27 L22: "For the remaining filters, delta_z_63 is either zero or decreases by 60 m because these filters act upon layers at higher altitudes". This is over ocean according to Table 2. You might want to describe what happens over land as well.

This sentence applies to land and ocean because the "remaining filters" are the isolated 80 km, CAD, and cirrus fringe filters. The text in red was added to clarify.

> *For the remaining filters, $\Delta z_{63}$ is zero or decreases by 60 m (over land and ocean) because these filters act upon layers at higher altitudes.*

Legend of Table 2: "and with no filters against all filters and with each filter applied independently". Consider rephrasing. Also, what is the reasoning behind the ocean and land separation in the Table?

Rephrased to:

> *Global metrics comparing changes in level 3 mean AOD when all filters are applied (top row) and when each filter is applied independently (remaining rows)*
* * *
Fig. 22: It is not clear why the authors show the surface elevation in this figure (not discussed in the text).

The following was added to P28 L16: *"Median surface elevations are shown to indicate altitudes where the number of samples averaged begins to decrease (often rapidly) relative to higher altitudes, thereby increasing the uncertainty in the mean values being compared."*
* * *
Fig. 23: Consider adding a mean AOD histogram as a fourth plot (instead of values on Fig 23a)

The mean values on Fig. 23a suite the analysis adequately by denoting which regions have higher AOD on average relative to others. Histograms would tell a similar story. However, adding 12 histograms onto the same

plot (one for each region) would be difficult to interpret compared to the simple mean values. I prefer the keep the simple mean values reported in Fig 23a in lieu of adding the AOD histogram as a fourth plot.
* * *
P30 L6: "greater geometric depth". Am I not understanding this correctly? Doesn't this mean z_63_filtered > z_63_non_filtered? Which would mean a higher altitude under which 63 % of the aerosols reside (instead of a geometric depth)?

Yes, your interpretation is correct. This means the altitude under which 63 % of the aerosol reside is higher after filtering. The "greater geometric depth" phrase assumes that the base altitude remains the same. In that case, greater geometric depth means the same thing as higher altitude. However, I prefer the language you suggest, so the text is reworded as:

> *The change in extinction scale height $\Delta z_{63}$ is positive for most regions (Fig. 23(b)), indicating that the altitude containing the bulk of the mean AOD is higher after quality filtering.*
* * *
P30 L16-18: "When aerosol loading is low, rejecting just a small number of aerosol samples may have a larger impact on sigma than in regions where aerosol loading is high since there are not many aerosol samples to begin with" This is not clear. Consider rephrasing.

Agreed. Rephrased to:

> *When aerosol loading is low, rejecting just a small number of aerosol samples may have a large impact on $\bar{\sigma}$ because there are not many aerosol samples to begin with.*
* * *
P31 L16: similar levels of uncertainty

Corrected
* * *
P31 L24: in order to reduce

Corrected
* * *
Appendix: Table B1 comes sooner in the text than appendix A: consider switching Appendix A and B.

Done. Appendices A and B are now switched. References to the tables/equations in the main text and supplementary material have been updated.
* * *
Fig. S1-S2 consider "quality screening" instead of "data filtering" to match the legend of Fig. 6-7
Done